# VTBench: Comprehensive Benchmark Suite Towards Real-World Virtual Try-on Models

## Abstract

While virtual try-on has achieved significant progress, evaluating these models towards real-world scenarios remains a challenge. A comprehensive benchmark is essential for three key reasons: (1) Current metrics inadequately reflect human perception, particularly in unpaired try-on settings; (2) Most existing test sets are limited to indoor scenarios, lacking complexity for real-world evaluation; and (3) An ideal system should guide future advancements in virtual try-on generation. To address these needs, we introduce the **V**irtual **T**ry-on **Bench**mark (**VTBench**), the first-ever hierarchical try-on benchmark suite that systematically decomposes virtual image try-on into hierarchical, disentangled dimensions, each equipped with tailored test sets and evaluation criteria. VTBench exhibits three key advantages: 1) Multi-Dimensional Evaluation Framework: The benchmark encompasses five critical dimensions for virtual try-on generation (*e.g.,* overall image quality, texture preservation, complex background consistency, cross-category plausibility, and hand-occlusion handling). Granular evaluation metrics of corresponding test sets pinpoint model capabilities and limitations across diverse, challenging scenarios. 2) Human Alignment: Human preference annotations are provided for each test set, ensuring the benchmark's alignment with perceptual quality across all evaluation dimensions. 3) Valuable Insights: Beyond standard indoor settings, we analyze model performance variations across dimensions and investigate the disparity between indoor and real-world try-on scenarios. To foster the field of virtual try-on towards challenging real-world scenarios, VTBench will be open-sourced, including all test sets, evaluation protocols, generated results, and human annotations.

## 1 Introduction

Image-based virtual try-on (VTON) (Dong et al., 2019; Ge et al., 2021a; Issenhuth et al., 2020; Han et al., 2019; 2018; He et al., 2022; Kim et al., 2024; Minar et al., 2020; Wang et al., 2018; Xie et al., 2023; Yang et al., 2020; 2024; Zhu et al., 2024) is a widely adopted and promising image synthesis technology in the e-commerce industry. Its primary goal is to enhance the shopping experience for consumers and minimize advertising expenses for clothing merchants. The VTON task involves generating an image of a human model wearing a specific garment. Over the past few years, researchers have dedicated significant efforts, such as Generative Adversarial Networks (Goodfellow et al., 2020) and Diffusion models (Rombach et al., 2022), to achieve more realistic and precise virtual try-on results. As image virtual try-on models advance, there is an urgent demand for robust evaluation. These evaluation methods must not only align with human perceptual judgments of generated try-on results but also provide reliable metrics for assessing model performance. Furthermore, they should reveal individual models' specific capabilities and limitations, offering actionable insights to guide future improvements in data selection, training strategies, and model architecture selection, which ultimately drives progress toward more sophisticated real-world applications.

Existing virtual try-on evaluations usually adopt the paired evaluation metrics and replace the garment-agnostic model with the target garment. But it fails to timely assess the performance of virtual try-on in unconstrained settings without the guidance of paired ground-truths. Moreover, acquiring paired data is inherently challenging, especially when attempting to test arbitrary clothing items on diverse models across varied real-world scenarios. The complexity makes such data collection exceptionally difficult. Consequently, there exist strong desires of unpaired evaluators that are universal and

effective to reliably reflect the quality of virtual try-on in open and challenging real-world scenarios getting rid of the strict paired shooting conditions.

Currently, existing metrics for image virtual try-on generation mainly consists of paired metrics, such as Structural Similarity (SSIM) (Wang et al., 2004) and Learned Perceptual Image Patch Similarity (LPIPS) (Zhang et al., 2018), and unpaired metrics, for example, Frechet Inception Distance (FID) (Parmar et al., 2022) and Kernel Inception Distance (KID) (Bińkowski et al., 2018). These unpaired metrics are inconsistent with human evaluations and also neglect the unique, challenging characteristics of the virtual try-on field in real-world scenes. Hence, there is a pressing need for an evaluation framework that aligns closely with human perception and is specifically designed for the characteristics of virtual try-on models. To this end, we introduce VTBench, the first comprehensive benchmark suite for evaluating virtual try-on model performance. VTBench has three appealing properties: 1) comprehensive evaluation dimensions and corresponding test sets catering for difficult try-on challenging cases, 2) Novel unpaired reliable evaluators highly close with human alignment, and 3) building the comprehensive and most recent SOTA baselines, and providing valuable insights.

First, our pipeline incorporates a hierarchical evaluation framework that systematically decomposes virtual try-on image quality through a structured and disentangled approach. The framework organizes the assessment into three fundamental dimensions: General Image Quality, Garment Preservation, and Auxiliary Consistency, each of which is subsequently refined into more specific evaluation criteria. This multi-level architecture ensures precise isolation and independent assessment of each quality aspect, eliminating potential interference between evaluation variables, as demonstrated in Fig. 1. For example, in terms of *Garment Preservation*, we disentangle this coarse dimension into the finer ones including the texture and cross-category plausibility to evaluate try-on model's capabilities in texture maintenance and cross-categories cases (*e.g.,* dress↔any). Each finer evaluator is supported by high-quality self-collected labor-consuming dataset. Specifically, to evaluate the ability of texture maintenance, we geniously convert texture judgments into font structure texture evaluations and collect the garments with different size of font. We carefully collect four test sets customized for corresponding four finer dimensions in the aspects of texture and size garment, and the auxiliary background and hand-occlusion consistency.

Secondly, each granular dimension is supported by novel unpaired specifically-customized metrics overcoming the considerable difficulty of paired try-on dataset collections. A total of four novel unpaired evaluators are formulated to comprehensively verify the model's capabilities in six granular dimension, breaking the great barriers of current try-on benchmark only focusing on the similarity and distribution gap. Moreover, we systematically demonstrate that our evaluators exhibit strong alignment with human perception across all fine-grained evaluation dimensions. Specifically, we use the image try-on models to sample synthetic try-on results, and then arrange human annotators to give users' preferences based on the six granular dimensions. We demonstrate that VTBench evaluations exhibit a strong correlation with human preferences.

Lastly, VTBench's multi-dimensional approach compared with previous single and limited try-on benchmarks can provide valuable insights to the virtual try-on community towards more complicated real-world scenarios. For example, through our hierarchical dimension system, we provide granular feedback on the capabilities and limitations of virtual try-on across diverse evaluation dimensions. This framework not only thoroughly assesses existing models but also illuminates the path toward training sophisticated try-on generation systems, offering guidance to refine architectures and data strategies for higher-quality outputs.

We hereby open-source VTBench in its entirety, comprising a comprehensive suite of evaluation dimensions, novel evaluation method, curated test collections, generated virtual try-on results, and annotated human preference datasets. The research community is cordially invited to benchmark their image-based virtual try-on generation models against this standardized evaluation framework.

## 2 RELATED WORKS

**Image Virtual Try-on Models.** Over the past decade, image-based virtual try-on (Cui et al., 2024) has garnered significant research attention, presenting itself as an increasingly promising field of remarkable e-commerce, which enhances the shopping experience for consumers. A series of studies based on Generative Adversarial Networks (GANs)(Ge et al., 2021b; Issenhuth et al., 2020; Lee

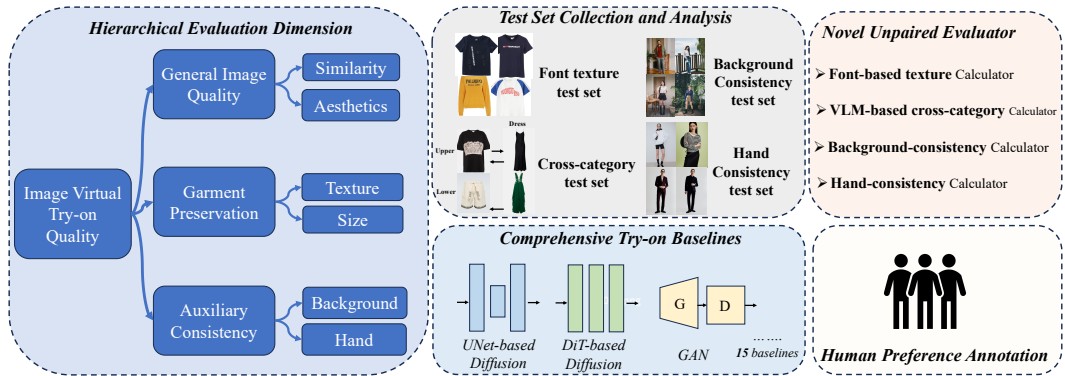

Figure 1: **Overview of VTBench**. We propose VTBench, the first comprehensive benchmark suite for evaluating image-based virtual try-on models. To enable fine-grained and objective assessment, we present a hierarchical Evaluation Dimension Suite that systematically decomposes "image virtual try-on quality" into multiple well-defined dimensions. For each dimension and content category, we curate dedicated test sets and develop robust metrics, then sample virtual try-on images from 16 models built on different foundations to provide in-depth insights. Furthermore, we conduct human preference annotations of the try-on results across all dimensions, demonstrating strong alignment between VTBench's automated evaluations and human perceptual judgments.

et al., 2022; Li et al., 2021; Xie et al., 2023; Yang et al., 2023; Chopra et al., 2021; Chen et al., 2023), UNet-based Diffusion (Chen et al., 2024b; Liang et al., 2025; Team, 2024; Zhu et al., 2023; Choi et al., 2024; Li et al., 2024), and DiT-based Diffusion structures (Luo et al., 2025; Jiang et al., 2024; Ni, 2025; Luan et al., 2025; Wan et al., 2024; Zheng et al., 2024) have emerged for high-fidelity and high-realism image virtual try-on. The exponential growth of virtual try-on generation models underscores the urgent need for standardized evaluation protocols that can both analyze existing capabilities and inform future research directions. In this context, VTBench emerges as a pioneering solution, providing a meticulously designed benchmark suite that facilitates hierarchical, human-aligned performance analysis towards more challenging real-world try-on scenarios.

**Evaluator of Virtual Try-on Generative Models.** Existing virtual try-on models typically use metrics as follows: In paired try-on settings in test datasets, the reconstruction accuracy is assessed by LPIPS (Zhang et al., 2018) and SSIM (Wang et al., 2004). In unpaired settings, only FID (Parmar et al., 2022) and KID (Bińkowski et al., 2018) are reported due to the absence of ground truth. The image similarity CLIP (Radford et al., 2021) is used in (Chen et al., 2024a). To our knowledge, we are the first to build a comprehensive benchmark and enrich more novel and reliable metrics.

**Comparison with VBench** (Huang et al., 2024). 1) Evaluation Scope: VBench focuses on general video generation, while VTBench is the first comprehensive benchmark specifically designed for VTON in real-world scenarios. 2) Metric Innovations: VBench presents a 16-dimensional framework for general video evaluation, while VTBench devises dedicated metrics for VTON-specific challenges, including cross-category plausibility, complex backgrounds, hand occlusion, and texture preservation. Moreover, considering the difficulty of collecting large-scale, diverse paired post-try-on images, VTBench systematically tackles these issues with tailored unpaired evaluation metrics.

## 3  VTBench: Virtual Try-on Benchmark

In this section, we detail the introduction of VTBench's components. Section 3.1 outlines the rationale behind designing the six evaluation dimensions, including their definitions and assessment evaluators. Section 3.2 elaborates on the test dataset collection process tailored for multidimensional analysis. Finally, in Section 3.3, we validate VTBench's alignment with human perception through dimension-specific preference annotations.

### 3.1  Hierarchical Evaluation Dimensions

We emphasize that the motivation of the VTbench suite stems from the thorough analysis of real-world virtual try-on (VTON) scenarios as follows:

**1.) Limitation of current VTON evaluators**: Existing metrics, such as FID and KID, measure only the distribution distance between image sets and are highly sensitive to data size. They fail to access single images and often misalign with human judgments.

**2.) Absence of multidimensional evaluation**: Existing VTON metrics reduce model performance to a single score, which oversimplifies the evaluation and fails to reveal specific strengths and weaknesses of models across diverse dimensions. This unidimensional approach limits actionable insights for improving model architectures and training strategies. Therefore, we propose a multidimensional evaluation framework that decomposes "virtual try-on performance" into distinct, well-defined components for granular analysis.

Specifically, we break down "virtual try-on quality" into six disentangled dimensions, each assessing a particular aspect of VTON generation. Considering the essence of virtual try-on tasks aims to modify the corresponding garment zones and keep other elements the same with the model image. Thus, at the top level, three main perspectives are considered: (1) General Image Quality, focusing on global perceptual fidelity and distribution alignment. (2) Garment Preservation, assessing the realistic substitution and retention of garment attributes, such as cross-category plausibility and texture. (3) Auxiliary Consistency, evaluating the stability of non-garment elements, including backgrounds and occluded regions, which VTON models often disrupt.

### 3.1.1 GENERAL IMAGE QUALITY

We split "General Image Quality" into two finer aspects, "Similarity" and "Aesthetics", where the former only considers the general distribution similarity between the ground-truths and synthetic images, and the latter assesses aesthetics and harmonization of global images. **1) Similarity**: We adopt the existing FID and KID to calculate the distribution distance of the deep feature space between model ground-truths and synthetic virtual try-on images. **2) Aesthetics**: The aesthetic score quantifies subjective experiences in virtual try-on results, including overall harmony (*e.g.,* integration between garments and human poses/backgrounds), color coordination, and style consistency, reflecting users' perception of "beauty." For instance, a virtually ill-fitting garment may be physically accurate yet receive a low aesthetic score due to unnatural wrinkles or mismatched lighting. Moreover, the aesthetic score captures higher-level visual flaws, such as the presence of dissonance and anatomical distortions (*e.g.,* unnatural body artifacts). To this end, we introduce aesthetic scores and calculate them based on the CLIP+MLP Aesthetic Score Predictor.

### 3.1.2 GARMENT PRESERVATION

Despite advancements of recent emergence fancy paradigms, virtual try-on still faces significant garment-fitting challenges, particularly in two areas: (1) **Shape-aware fitting**, where clothing information leakage occurs in cross-category or shape-mismatched try-on scenarios, leading to generated garments that cover the entire agnostic mask region; and (2) **Rich texture-aware maintenance**, where the transformation of intricate textures (*e.g.,* patterns, text, stripes, trademarks) to the target model is hindered by the limitations of current structures and training manners. To systematically assess model performance and robustness in addressing these challenges, we decompose "Garment Preservation" into two key dimensions, namely Cross-Category Plausibility and Texture Fidelity, and develop corresponding evaluation metrics that align with human perceptual judgments.

**Cross-Category Plausibility**: Virtual try-on methods often handle simple scenarios (*e.g.,* long sleeves ↔ short sleeves) but struggle with cross-category transfers such as long skirts ↔ upper jackets. The core difficulty is the absence of logical reasoning about how a target garment should be placed on the model, which we formalize by partitioning the image into three zones: a reconstruction zone that learns from the model image, a try-on zone that references the garment image, and an imagination zone that leverages diffusion-based generative capacity. To quantify cross-category try-on ability strongly needed by real users and personalized styling demands, we introduce Vision Language Models (VLMs) to analyze the logical consistency of garment placement. Given a model image, target garment, and the virtual try-on result, we employ Qwen-VL-Max to determine whether the result is shape-fitting and logically consistent for cross-category transfers. The $\text{VLM}_s$ metric is formultated as:

$$E_{size} = \frac{S_g[\text{VLM}(\text{Concat}(I_m, I_g, I_{syn}))]}{S_{\text{cases}}} \tag{1}$$

where $S_g$ is the number of reasonable cross-category cases, $S_{cases}$ is the total cases. The model image $I_m$, garment image $I_g$, and synthetic try-on image $I_{syn}$ are concatenated as the inputs to the Qwen-VL-Max (Bai et al., 2025) model.

**Texture Fidelity**: We identify that fine-grained texture transfer and consistent clothing appearance are crucial aspects of virtual try-on, yet there is a lack of suitable evaluation metrics in this domain. Existing metrics like SSIM (Wang et al., 2004) have significant limitations. For example, even if the flowers on the target garment and the generated image are visually identical, a slight positional deviation can lead to a low metric score. Thus, we propose a font texture similarity (FTS) metric to evaluate the capability of font texture maintenance. Specifically, we introduce the OCR model (Li et al., 2022) for text feature extraction, and then use these features to customize a font-texture similarity metric to represent the capability of texture maintenance. Specifically, the text regions in both the clothing image and the try-on result image can be obtained. Then we calculate the text similarity for each corresponding region via optimizing language processing distance criteria consisting of term frequency similarity and edit distance-based similarity components. Specifically, the component of edit distance-based similarity is:

$$E_{edit}(\boldsymbol{s}_{gt}, \boldsymbol{s}_p) = 1 - \frac{\text{LEV}(\boldsymbol{s}_{gt}, \boldsymbol{s}_p)}{max(|\boldsymbol{s_{gt}}|, |\boldsymbol{s_p}|)} \tag{2}$$

where $s_{gt}$ is the text output of PPOCRv3 model from the target garment, and $s_p$ is the text output of PPOCRv3 model from the synthesis try-on image. LEV denotes the Levenshtein distance from the string $s_p$ to the string $s_{gt}$. The part of term frequency similarity is shown as:

$$E_{term}(\boldsymbol{s}_{gt}, \boldsymbol{s}_p) = \frac{\text{TF}(\boldsymbol{s}_{gt}) \cdot \text{TF}(\boldsymbol{s}_p)}{||\text{TF}(\boldsymbol{s_{gt}})|| \times ||\text{TF}(\boldsymbol{s_p})||} \tag{3}$$

where TF is the term frequency calculation, and $E_{term}$ calculates the Cosine Similarity of the term frequency between the string $s_{gt}$ and $s_p$. The true positive TP, false positive FP, and false negative FN predictions of all font characters are listed as $\text{TP}_s = |s_{gt}| \cap |s_p|$, $\text{FP}_s = |s_p| - |s_{gt}|$, and $\text{FN}_s = |s_{gt}| - |s_p|$. $||$ means the recognized font character code. Precision $E_p$, Recall $E_r$, and F1-Measure $E_f$ are calculated to analyze the accuracy of font character recognition.

Our proposed text semantic-texture similarity is finally formulated as:

$$E_{TSS} = w_1 * \frac{1}{N} \sum_{i=1}^{N} E_{term}^i + w_2 * \frac{1}{N} \sum_{i=1}^{N} E_{edit}^i + w_3 * \frac{1}{N} \sum_{i=1}^{N} (E_p^i + E_r^i + E_f^i) \tag{4}$$

where the weights of $w_1$, $w_2$ and $w_3$ are set to 0.2. $i$ and $N$ are the $i$-th region and the total number of regions, respectively. The average similarity score across multiple regions is considered as the final similarity score.

Inspired by MagicBrush (Zhang et al., 2023), we further assess the preservation of texture features before and after the virtual try-on process from two perspectives. At the visual texture level, we calculate the cosine similarity between the embedding of the garment image $I_g$ and the embedding of the cropped garment region of the synthesized try-on image $I_{syn}$. This process is formulated as:

$$E_{\mathcal{E}} = \frac{\mathcal{E}(I_g) \cdot \mathcal{E}(\text{Crop}(I_{syn}, \text{bbox}_g))}{||\mathcal{E}(I_g)|| \times ||\mathcal{E}(\text{Crop}(I_{syn}, \text{bbox}_g))||}, \tag{5}$$

where $\mathcal{E}$ denotes a pretrained image encoder (*e.g.,* CLIP, DINO).

For image-text feature assessment, we use the garment image to construct a precise textual description $T_{syn}$ for the synthetic try-on image, and subsequently employ CLIP-T to evaluate the similarity between the description and the synthetic try-on image.

Additionally, following the aforementioned Cross-Category Plausibility protocol, we utilize Qwen-VL-Max to comprehensively evaluate the synthetic try-on image across multiple aspects—including semantic alignment with garment textual prompts, texture fidelity, color consistency, and representation of wrinkles—yielding the overall evaluation metric $\text{VLM}_t$.

### 3.1.3 AUXILIARY CONSISTENCY

The fundamental objective of virtual try-on methodologies lies in accurately identifying modifiable regions while preserving invariant areas within the model image. The aforementioned garment

evaluation metrics specifically assess the visual quality of synthesized garments in the target region. This section systematically evaluates two critical aspects: preservation consistency of unmodified regions, and contextual coherence in complex scenarios. Real-world virtual try-on applications typically involve cluttered backgrounds with frequent human-object interactions, presenting significant challenges for background preservation. Furthermore, hand-occluded garment scenarios, which frequently occur in real-world applications involving complex human poses, introduce additional complications for garment synthesis algorithms. To comprehensively evaluate model performance and robustness in addressing consistency challenges, we mainly disentangle "Auxiliary Consistency" into two aspects including Background Consistency and Hand Consistency. For each dimension, we develop quantitative evaluation metrics that are both computationally rigorous and perceptually aligned with human visual assessment.

**Background Consistency**: In virtual try-on for real-world scenarios, the background is often highly complex, involving interactions between people and objects. However, the current inpainting-based virtual try-on paradigm tends to alter background information to some extent. As an ideal garment replacement algorithm, it should minimize modifications to reasonable areas while preserving background consistency as much as possible. Additionally, any altered background regions should maintain harmony with the surrounding context. Then we respectively calculate the pixel-wise and semantic-wise similarity after determining the target mask zone. $M_{GT}$ and $M_{syn}$ are the human mask of model images and synthetic try-on results, respectively. We dilate the maximum zone and obtain the residual mask zone $M_D$ around human as follows:

$$M_D = \text{Dilate}[(M_{GT}, M_{syn})_{\max}] - M_{GT} \tag{6}$$

Afterwards, the pixel-level of background consistency is calculated based on L1 distance around $M_D$ zone as follows:

$$E_{pixel} = |I_{syn} - I_m|_{\text{L1}, M_D} \tag{7}$$

Besides, the semantic-wise similarity of background consistency is obtained based on DINO model around $M_D$ residual area.

$$E_{semantic} = |I_{syn} - I_m|_{\text{DINO}, M_D} \tag{8}$$

**Hand-Structure Consistency**: Current virtual try-on systems typically require models to maintain rigid poses with hands positioned strictly at their sides, ensuring garments remain completely unobstructed. This artificial constraint significantly limits the practical applicability of existing methods in real-world scenarios. The challenge is particularly severe in digital human applications, where natural poses frequently involve complex hand gesture language that occluded garment. Such occlusions contribute a lot of failure cases for contemporary virtual try-on algorithms. It requires high-accommodation of different pose variations while maintain the hand appearance and structure consistent especially on hand-occluded cases. We introduce the joint position error (JPE) to get the each joint distance between the synthetic and model images as the final error:

$$E_{\text{MPJPE}} = \frac{1}{N_s} \sum_{i=1}^{N_s} \|m_{syn,s}(i) - m_{model,s}(i)\|_2 \tag{9}$$

where $m_{syn,s}(i)$ and $m_{model,s}(i)$ are the functions that return the coordinates of the i-th joint of skeleton from the synthetic try-on images and the model images, respectively. $N_s$ is the number of the joints in skeleton S.

## 3.2 TEST DATASET COLLECTION AND ANALYSIS

As an evaluator suite of each hierarchical dimension, it not only requires the custom-designed metrics but only cleaning the whole dataset to focus on the specific dimension which requiring greatly intensive labor annotations. Specifically, we collect an additional dataset of 50,000 images sourced from online retail sites. Considering just varying the specific dimension without involving other dimension disturbance, we carefully filter the whole dataset and classify the remaining dateset into the four classes catering for each well-disengaged evaluation dimension.

**Complex Background Consistency Dataset (CBC)**: The significance of complex background testing is to transition garment virtual-try-on algorithms from "controlled lab environments" to "real-world deployment scenarios." Only by passing rigorous validation in complex settings can virtual try-on

Table 1: VTBench quantitative results of 16 SOTA methods on different network paradigms on our benchmark datasets of each dimension. The best results of all baselines are highlighted in **underline**, and the best results of each specific framework (*i.e.,* GAN, UNet-based or DiT-based Diffusion model) is marked in **bold**.

| Baseline Models | Image Virtual Try-on Quality | | | Garment Preservation | | | | | | Auxiliary Consistency | | Overall |
| | Similarity | | Aesthetics | | | Texture | | | Cross-Category | Background | Hand-Structure | Rank |
| | FID↓ | KID↓ | $A_s$↑ | $F_t$↑ | VLM$_t$↑ | Clip-I↑ | Dino↑ | Clip-T↑ | VLM$_s$↑ | $B_{consist}$↓ | $H_{consist}$↓ | $R_{avg}$↓ |
| *GAN-based* | | | | | | | | | | | | |
| 2021 PF-AFN | 74.7482 | 40.8694 | 3.7881 | 0.3278 | 0.6840 | 0.7723 | 0.6551 | **0.3195** | - | 4109.3963 | 75.4443 | 12.1 |
| 2022 FS-VTON | 71.0491 | 40.7414 | 3.9055 | 0.3165 | 0.6871 | 0.7766 | 0.6681 | 0.3117 | - | 2940.1240 | 42.8215 | 11.3 |
| 2022 DA-FLOW | 100.2046 | 64.2756 | 3.8198 | 0.2690 | 0.5617 | 0.7466 | 0.5824 | 0.2929 | - | **2686.7483** | 179.8285 | 13.9 |
| 2022 HR-VITON | 56.5964 | 23.1358 | 4.5128 | 0.4099 | 0.6671 | 0.7781 | 0.6321 | 0.3037 | - | 6495.7036 | 43.4961 | 11.8 |
| 2024 SD-VITON | **49.6304** | **16.5367** | **4.5716** | **0.4786** | **0.7464** | **0.8194** | **0.7293** | 0.3112 | - | 6070.9095 | **38.6793** | **9.2** |
| *UNet-based Diffusion* | | | | | | | | | | | | |
| 2023 LaDI-VTON | 31.4483 | 7.1166 | 5.0518 | 0.0601 | 0.6850 | 0.7239 | 0.6483 | 0.2771 | - | 485.8788 | 45.7704 | 11.1 |
| 2024 StableVTON | **27.8368** | **2.2319** | 4.9862 | 0.2009 | 0.6868 | 0.8017 | 0.7011 | 0.3015 | - | **132.7260** | 40.2104 | 7.5 |
| 2024 TPD | 32.6199 | 6.7273 | 4.8379 | 0.3383 | 0.8560 | 0.8197 | 0.6935 | **0.3180** | - | 161.3437 | 39.4060 | 7.6 |
| 2024 CAT-DM | 28.7471 | 4.4718 | 4.9188 | 0.0140 | 0.6964 | 0.7211 | 0.6515 | 0.2725 | 12.8571 | 154.1549 | 42.4447 | 9.6 |
| 2024 IDM-VTON | 29.8710 | 4.9117 | **5.1243** | 0.3387 | 0.8439 | 0.8390 | **0.7469** | 0.3165 | 15.7143 | 173.7212 | 48.3768 | 6.2 |
| 2024 OOTD | 45.9138 | 16.7691 | 5.0367 | 0.2153 | 0.6021 | 0.7319 | 0.4905 | 0.2699 | 13.5714 | 205.4843 | 62.5795 | 11.9 |
| 2024 CatVTON | 29.2677 | 3.4891 | 5.0336 | **0.5433** | **0.8650** | 0.8425 | 0.7168 | 0.3175 | 21.4286 | 176.5129 | 35.7663 | 5.1 |
| 2025 VTON-HandFit | 28.9344 | 2.4419 | 5.1170 | 0.4949 | 0.8459 | **0.8443** | 0.7028 | 0.3180 | **29.0476** | 319.8210 | **15.4596** | **4.4** |
| *DiT-based Diffusion* | | | | | | | | | | | | |
| 2024 FitDit | **29.6194** | **3.6564** | **5.1135** | **0.6481** | **0.9065** | **0.8705** | **0.7660** | **0.3267** | **55.2381** | **137.4822** | 33.7351 | **2.1** |
| 2025 CrossVTON | 32.0386 | 3.9203 | 5.1018 | 0.5728 | 0.8950 | 0.8611 | 0.7418 | 0.3255 | 48.0952 | 715.6857 | **25.2247** | 4.1 |
| 2025 Leffa | 41.5223 | 5.3442 | 5.0987 | 0.6338 | 0.8919 | 0.8607 | 0.7377 | 0.3259 | 43.0952 | 212.5107 | 107.1843 | 5.7 |

technology genuinely serve daily life and evolve into a reliable 'digital fitting room' for consuming users. We filter these data into the complex scenarios including the street scene, wild scene, and none-simple indoor scene, and collect 5,00 images for further prepossessing and evaluation.

**Font Texture Fidelity Dataset (FTF)**: Font Texture Fidelity Dataset focuses on evaluating the model's ability to accurately transfer fine-grained textures of the font. In the FTF set, we observe that text structures serve as reliable indicators of the model's fine-grained texture transfer capabilities. We manually annotate and collect 600 images for further prepossessing and evaluations.

**Cross-Category Plausibility (CCP) Dataset**: To comprehensively evaluate cross-category virtual try-on performance, we construct the Cross-Category Plausibility Dataset (CCP) by systematically combining models and garments from four distinct categories in DressCode: top, lower, dresses, skirts. The dataset contains 400 base model-garment pairs with 100 pair category), which are strategically combined to create 1,000 test pairs that rigorously assess cross-category compatibility.

**Hand-occluded Consistency Dataset (HOC)**: Current virtual try-on datasets predominantly feature hands separated from the body, resulting in a scarcity of data for arbitrary hand poses and challenging hand-occlusion scenarios. To address this critical gap in real-world applications, we introduce a novel dataset focusing on complex hand poses that occlude garments. Our pipeline begins by processing images using a parsing model alongside the HaMeR model (Pavlakos et al., 2024) to detect hand bounding boxes and identify occlusion instances. Rigorous manual inspection was conducted to eliminate erroneous or low-quality entries, ensuring the dataset's reliability. From an initial pool of 50,000 images collected from online retail platforms, we curated the Hand-occluded Consistency Dataset (HOC), a specialized test set comprising 1,433 high-quality images. This dataset enables targeted evaluation of virtual try-on methods, specifically their robustness in handling occlusions.

### 3.3 HUMAN PREFERENCE ANNOTATION

We conduct large-scale human preference labeling on try-on results to verify the alignment between our evaluation metrics and human perception across the six evaluation dimensions. Given a garment $g_i$, and six image virtual try-on models $\{M_1, .., M_6\}$, each model group yields fifteen binary comparisons, for which annotators indicate their preference (*i.e.,* A or B is better or equal). Each dimension takes 20 samples, and total it provides $N \times 5 \times 16$ image comparisons. Human annotators follow the rules that only consider the specific evaluation dimension without disturbances of other irrelevant dimension.

## 4 EXPERIMENTS

We introduce the most-recent 16 baselines including GAN-based (PF-AFN (Ge et al., 2021b), FS-VTON (He et al., 2022), DA-FLOW (Bai et al., 2022), HR-VITON (Lee et al., 2022), SD-VITON (Shim et al., 2024)), U-Net-based Diffusion (LaDI-VTON (Morelli et al., 2023), StableVTON (Kim et al., 2024), TPD (Yang et al., 2024), CAT-DM (Zeng et al., 2024), IDM-VTON (Choi et al., 2024),

OOTD (Xu et al., 2024), CatVTON (Chong et al., 2024), VTON-HandFit (Liang et al., 2025)), and DiT-based Diffusion models (FitDit (Jiang et al., 2024), CrossVTON (Luo et al., 2025), Leffa (Zhou et al., 2025)) for VTBench evaluations. For a fair comparison, all results are reproduced by official open-source models with the recommended setting.

## 4.1 QUANTITATIVE RESULTS

In Tab. 1 and Fig. 2, we group all baselines by network architecture: GAN-based, UNet-based, and DiT-based Diffusion. The UNet-based diffusion model (StableVTON) achieves comparable similarity, aesthetic, and background consistency scores to the DiT-based model (FitDit), both significantly outperforming the GAN-based baseline (SD-VITON). These results demonstrate the superiority of Diffusion-based methods over GAN-based methods on the high-fidelity image generation capability bridging the gap between the synthetic and real image. FitDit, a DiT-architected diffusion model, outperforms GAN and UNet-based diffusion frameworks by notable margins in maintaining garment structural integrity, particularly in texture detail conservation and cross-category plausibility.

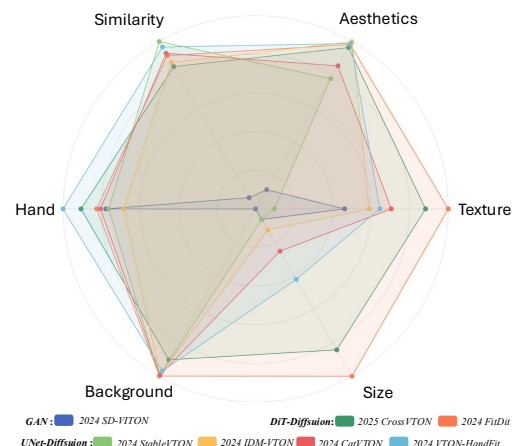

Figure 2: VTBench Evaluation Results of SOTA Virtual try-on Models including GAN, UNet-based and DiT-based Diffusion.

## 4.2 VALIDATING HUMAN ALIGNMENT

To verify that our evaluation method accurately captures human perceptual judgments, we conducted a large-scale human annotation study for each dimension. The correlation between VTBench evaluation results and human preference annotations is illustrated in Fig. 3.

**Win Ratio:** Following the VBench (Huang et al., 2024) video quality benchmark framework, we compute each model's win ratio based on human annotations. In pairwise comparisons: 1). A model receives 1 point if its output is preferred over the other's, while the competing model receives 0. 2). In case of a tie, both models are awarded 0.5 points. The win ratio for each model is then derived by dividing its total score by the number of pairwise comparisons it participated in.

**Per-Dimension Evaluation Analysis:** For each evaluation dimension, we compute model win ratios using both (1) VTBench automated evaluations and (2) human annotation results, then analyze their correlation in Fig. 3. Our findings demonstrate **strong alignment** between VTBench's per-dimension assessments and human preference judgments. Additionally, we provide alignment evaluations between other dimensions in Fig. 8 and Fig. 9. Notably, aesthetic scores show low correlation with human preference, suggesting that existing aesthetic metrics are inadequate for unpaired VTON tasks.

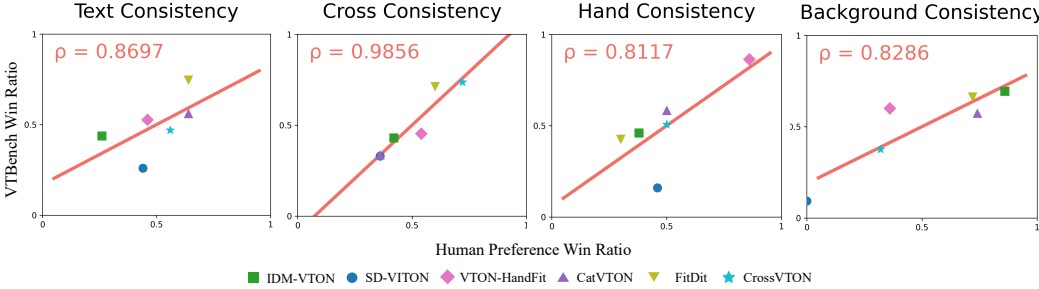

Figure 3: **Validate VTBench's Human Alignment**. Our experimental results demonstrate that VTBench evaluations across all dimensions exhibit a strong alignment with human perceptual judgments. Each plot illustrates the verification results for a specific VTBench dimension, where a single dot represents the human preference win rate (x-axis) and the VTBench evaluation win rate (y-axis) for a given virtual try-on generation model. To assess the correlation, we perform a linear regression analysis and compute the Spearman's rank correlation coefficient for each dimension.

**Other Visual Texture Metric Evaluation Analysis:** As demonstrated in Fig. 4 , we rigorously evaluate the efficacy of high-frequency variants of LPIPS and SigLIP for visual texture assessment in virtual try-on (VTON) scenarios. Our analysis reveals that LPIPS exhibits low correlation with human annotations, whereas SigLIP demonstrates significantly stronger alignment. This indicates that while SigLIP possesses non-trivial capability in measuring texture fidelity, its primary sensitivity lies in capturing high-level semantic consistency rather than low-level textural details. Furthermore, we assess two no-reference image quality metrics (blur estimation and noise estimation), which similarly show weak correlation with human judgments ($\rho < 0.25$ for both). These empirical findings substantiate the exclusion of conventional no-reference metrics from evaluation frameworks for unpaired VTON tasks, as they fail to capture perceptually meaningful distortions inherent in garment synthesis. This observation aligns with recent critiques regarding the inadequacy of standard image quality metrics for generative vision tasks. We leverage the Vision-Language Model (VLM) to assess two key aspects of skin tone preservation in synthetic try-on images: (1) the consistency of hand skin tone before and after visual try-on and (2) the chromatic coherence between hand skin tone and other visible body regions in the generated image. To validate the VLM's judgments, we conduct an additional human preference annotation study focused specifically on skin tone fidelity in synthetic try-on outputs. Our results reveal a high degree of agreement between VLM-generated scores and human annotations ($\rho > 0.85$ ), demonstrating that the VLM effectively captures human-perceived plausibility of skin tone continuity. Consequently, we formally integrate hand skin tone consistency as a core evaluation dimension within VTBench. This enhances its capacity to measure fine-grained visual realism by extending evaluation beyond structural alignment or textural fidelity to include perceptually critical chromatic cues. As shown in Fig. 6 , we validate the low correlation between FID/KID metrics and human preference ratings. We partition human preference-annotated try-on results into two groups, "Good" and "Bad", based on perceptual quality ratings. We then compute FID/KID scores between each group and the original ground truth person images. The FID/KID scores exhibit a counter-intuitive trend: the 'good' group yields higher (worse) scores than the 'bad' group. This quantitative analysis demonstrates that generic fidelity metrics do not reliably reflect human perceptual judgments in VTON tasks.

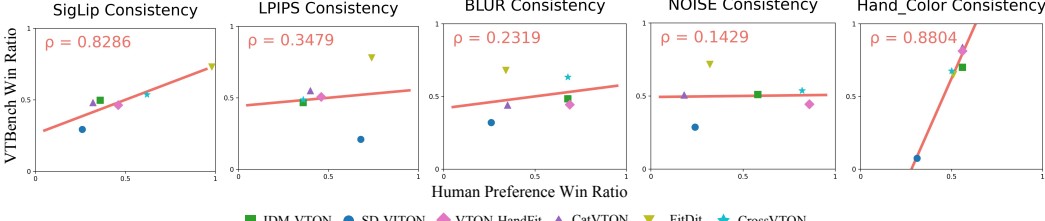

Figure 4: Validation of Additional Metrics with Human Annotations.

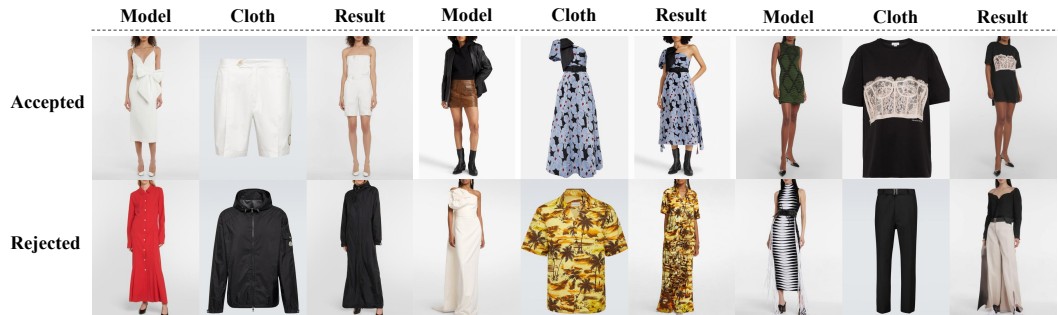

Figure 5: Accepted and Rejected VLM Response Cases in Cross-Category VTON.

## 5 INSIGHTS AND DISCUSSIONS

***DiT outperforms others on garment preservation:*** UNet's down-/up-sampling preserves global context at the cost of high-resolution latent features. In SD v1.5, the attention-related parameters allocated to latent resolutions above $64 \times 48$ comprise only 16% (Jiang et al., 2024), which impairs preservation of fine garment textures critical for high-fidelity virtual try-on. In contrast, SD3 removes

down-/up-sampling and concentrates attention parameters at $64 \times 48$ resolution (over 99%), thereby better maintaining high-resolution detail; consequently, DiT-based architectures are better suited for virtual try-on tasks that demand texture and detail fidelity.

***Similarity dimension fails to represent texture maintenance and cross-category plausibility:*** Similarity metrics such as FID and KID, which quantify distributional agreement between image batches, do not reliably reflect garment texture preservation or cross-category plausibility. As shown in Tab. 1, StableVTON attains the best FID/KID but scores substantially worse on texture preservation than FitDit, and VTON-HandFit achieves superior FID/KID while markedly underperforming in garment size preservation. These discrepancies indicate that FID and KID are insensitive to the texture- and geometry-specific aspects critical for virtual try-on. Therefore, we propose our garment texture and cross-category plausibility metric to correctly evaluate garment preservation of virtual try-on.

***More attention on garment preservation metrics:*** As a virtual try-on task, it is the absence of reasonable unpaired metrics to evaluate garment preservation, mainly consisting of texture and cross-category plausibility. When paired data are available, prior works typically evaluate by replacing the model's clothing with the ground-truth target garment. However, the limited availability and collection difficulty of paired try-on data constrain timely evaluation on out-of-domain, complex real-world scenarios. Above human alignment analysis verifies the effectiveness of proposed garment preservation metrics, and we call on that more attention should be assigned to these garment preservation metrics to judge garment-relevant texture quality and cross-category plausibility.

***GANs exhibit limitations in maintaining consistent background:*** As shown in Tab. 1, GAN-based methods perform much worse than diffusion-based methods in the pixel-wise background consistency. Diffusion models typically formulate virtual try-on as exemplar-based image inpainting and consequently exhibit strong, coherent inpainting behavior. GAN pipelines also generate and fuse try-on conditions (e.g., clothing-agnostic image, warped garment, pose map) and treat synthesis as an inpainting-like task, but their inpainting is less consistent. We attribute this gap to a foundation-model discrepancy: diffusion models produce more robust and harmonious inpainting, yielding more faithful foreground–background transitions. Finally, background consistency from the DiT-based Stable Diffusion 3 is comparable to that of UNet-based Stable Diffusion variants, indicating that replacing UNet with DiT in the diffusion backbone does not substantially improve background consistency.

***Hand-priors benefit the hand-occluded reconstruction:*** As shown in Tab. 1, VTON-HandFit attains the best hand-consistency score, reducing hand-consistency error by 38.7% relative to the second-best FitDit. VTON-HandFit adopts the encoding of structure-parametric and visual-appearance priors to solve hand pose occlusion problems. From Tab. 1, such hand prior embedding is verified to be effective in reconstructing the occluded hand in virtual try-on tasks. We recall that this hand-prior embedding probably facilitates the virtual try-on results, especially on hand-occluded scenarios.

## 6 CONCLUSION

As the virtual try-on generation garners increasing attention, a systematic and rigorous evaluation framework is crucial to assess the progress of existing models and steer future research directions. To address this need, we propose VTBench, the first comprehensive benchmark suite designed specifically for evaluating virtual try-on generation models. VTBench is characterized by its multi-dimensional evaluation criteria, human-aligned assessment methodology, and insight-rich analysis, making it a powerful tool for benchmarking current and future advancements in the field.

**Broader impacts and Limitations:** VTBench will serve as a foundational resource for researchers, facilitating more objective comparisons and inspiring further innovation in virtual try-on generation. We believe this work represents a significant contribution to both the virtual try-on generation community and the broader field of AI-driven fashion technology. But VTBench currently focuses exclusively on the most prevalent image virtual try-on task. It does not support video-based virtual try-on evaluations, particularly in assessing temporal smoothness.

## 7 REPRODUCIBILITY STATEMENT

We have already elaborated on all the models or algorithms proposed, experimental configurations, and benchmarks used in the experiments in the main body or appendix of this paper. Furthermore, we declare that the entire code used in this work will be released after acceptance.

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

# A APPENDIX

## A.1 THE USE OF LARGE LANGUAGE MODELS

We use large language models solely for polishing our writing, and we have conducted a careful check, taking full responsibility for all content in this work.

## A.2 ETHICS STATEMENT

This study adheres to the ethical guidelines outlined by ICLR. All data used in this work are collected exclusively from public domains and open-licensed sources. We ensured that no private, sensitive, or personally identifiable information is present in the dataset, and annotation procedures were designed to uphold principles of privacy and fairness throughout. To encourage responsible research and reproducibility in Virtual Try-on Benchmark, we commit to publicly releasing both the dataset and code strictly for academic purposes. Our efforts are intended to contribute to the transparency and integrity of the research community.

## A.3 ADDITIONALLY RESULTS

In this supplementary document, we provide additional results to complement our main paper as follows:

• We present more radar charts of our benchmark to explicitly demonstrate each baselines' superiority and limitation on all dimensions testified on the baselines including GAN, UNet-based and DiT-based Diffusion models, as shown in Fig. 7.

• We also conduct human preference labeling on the Aesthetics metric, which verifying the limitation of Aesthetics metric which demonstrate weak alignment with human perception in Fig. 8.

• We provide more qualitative comparisons with state-of-the-art models, respectively on Cross-category Plausibility Dataset, Complex Background Consistency Dataset (CBC), Hand-occluded Consistency Dataset (HOC), and Texture Fidelity Dataset (FTF), as shown in Fig. 10 to 14.

• We offer a preview of four labor-consuming self-collected dataset catering for each well-disengaged evaluation dimension, including Complex Background Consistency Dataset (CBC) in 15, Texture Fidelity Dataset (FTF) in 17, and Hand-occluded Consistency Dataset (HOC) in 16.

Data Usage & Copyright Protection: All image data utilized in this project has been properly authorized, with necessary permissions obtained. To prevent misuse, all generated images will be stamped with visible watermarks.

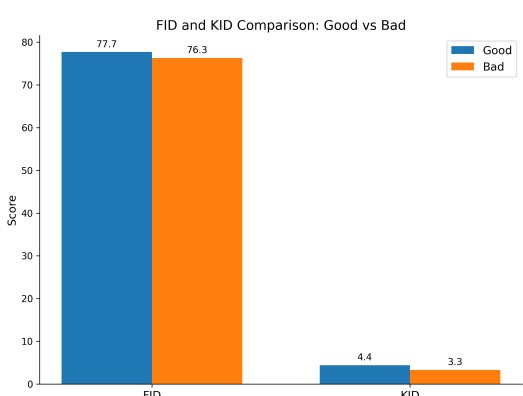

Figure 6: KID/FID Comparison "Good" vs "Bad" Human Ratings.

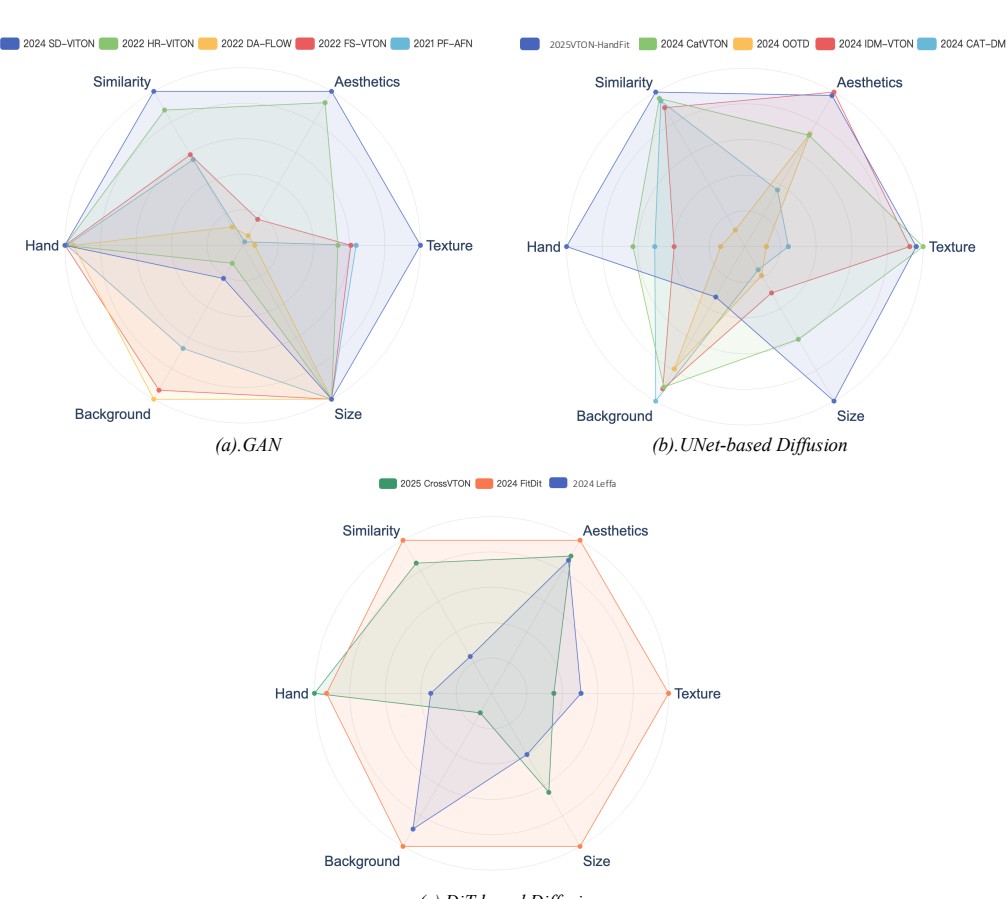

Figure 7: VTBench Evaluation Results of SOTA Virtual try-on Models including GAN, UNet-based and DiT-based Diffusion.

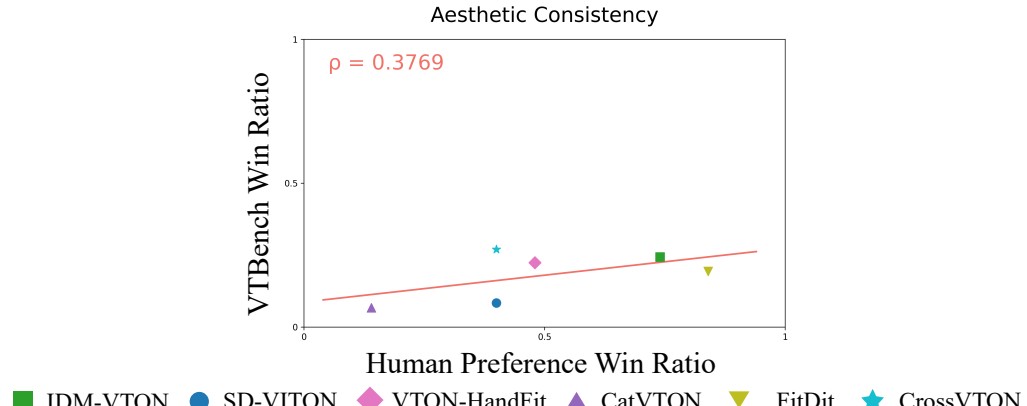

Figure 8: **Validate VTBench's Human Alignment with Aesthetics socre**. Our experimental results demonstrate that aesthetics evaluation exhibit a weak alignment with human perceptual judgments. The plot illustrates the verification results for a specific VTBench dimension, where a single dot represents the human preference win rate (x-axis) and the VTBench evaluation win rate (y-axis) for a given virtual try-on generation model. To assess the correlation, we perform a linear regression analysis and compute the Spearman's rank correlation coefficient.

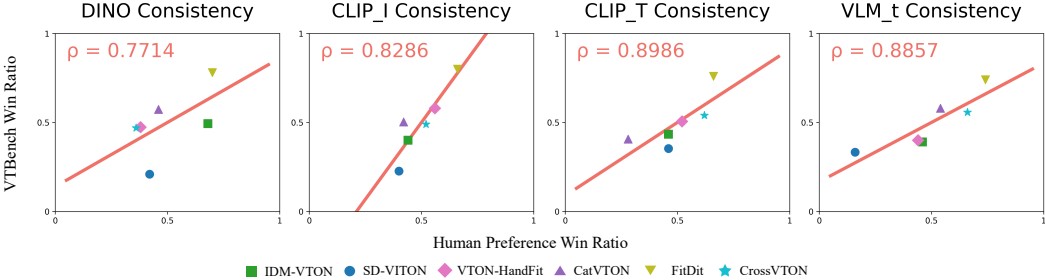

Figure 9: **Validate VTBench's Human Alignment with fine-grained texture scores.** Our experimental results demonstrate that fine-grained texture evaluations (DINO, CLIP-I, CLIP-T, $VLM_t$) exhibit a strong alignment with human perceptual judgments. Each plot illustrates the verification results for a specific VTBench dimension, where a single dot represents the human preference win rate (x-axis) and the VTBench evaluation win rate (y-axis) for a given virtual try-on generation model. To assess the correlation, we perform a linear regression analysis and compute the Spearman's rank correlation coefficient for each dimension.

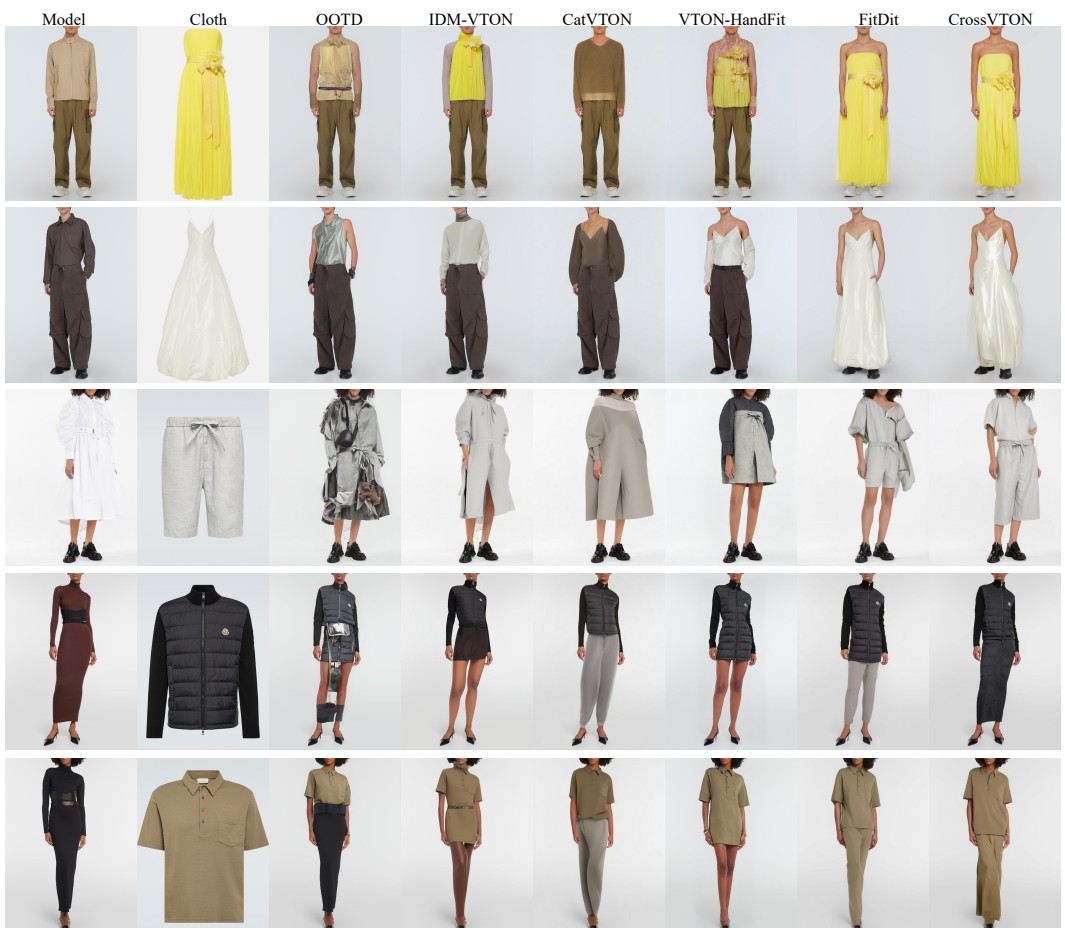

Figure 10: Visual results on Cross-Category Plausibility (CCP) Dataset to analyze the cross-categories virtual try-on ability.

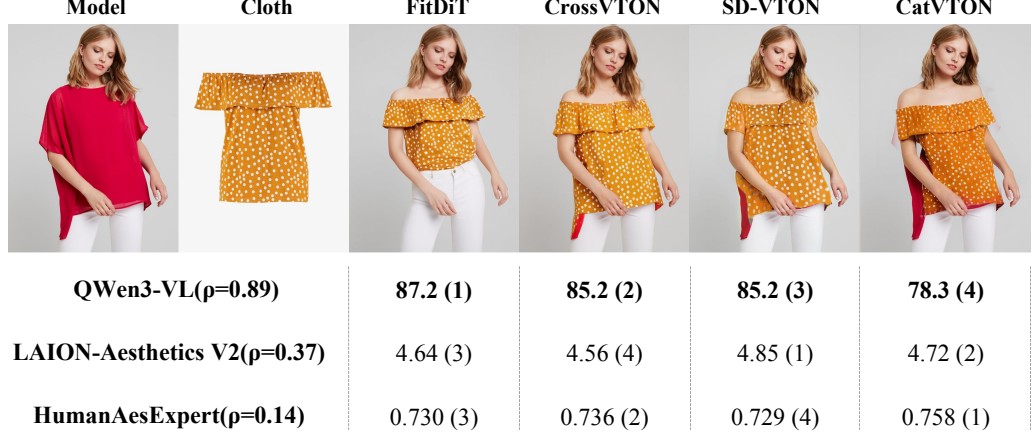

| | Model | Cloth | FitDiT | CrossVTON | SD-VTON | CatVTON |
|---|---|---|---|---|---|---|
| QWen3-VL(ρ=0.89) | | | 87.2 (1) | 85.2 (2) | 85.2 (3) | 78.3 (4) |
| LAION-Aesthetics V2(ρ=0.37) | | | 4.64 (3) | 4.56 (4) | 4.85 (1) | 4.72 (2) |
| HumanAesExpert(ρ=0.14) | | | 0.730 (3) | 0.736 (2) | 0.729 (4) | 0.758 (1) |

Figure 11: Example of Aesthetic Quality Scores from Different Assessment Metrics for Virtual Try-on Methods, where numbers in parentheses indicate rank (1=best, 4=worst).

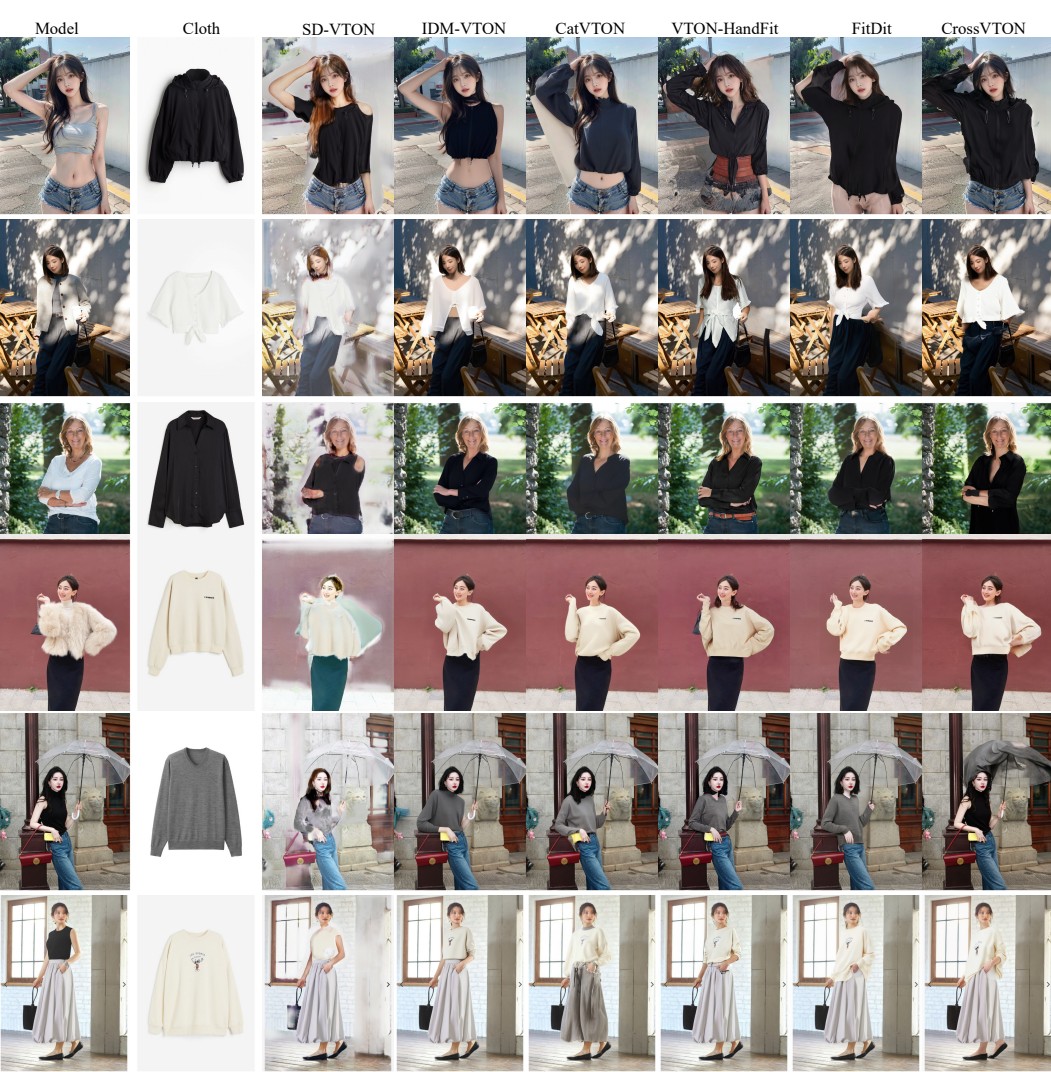

Figure 12: Visual results on Complex Background Consistency Dataset (CBC) to analyze the complex background consistency maintenance ability for virtual try-on tasks.

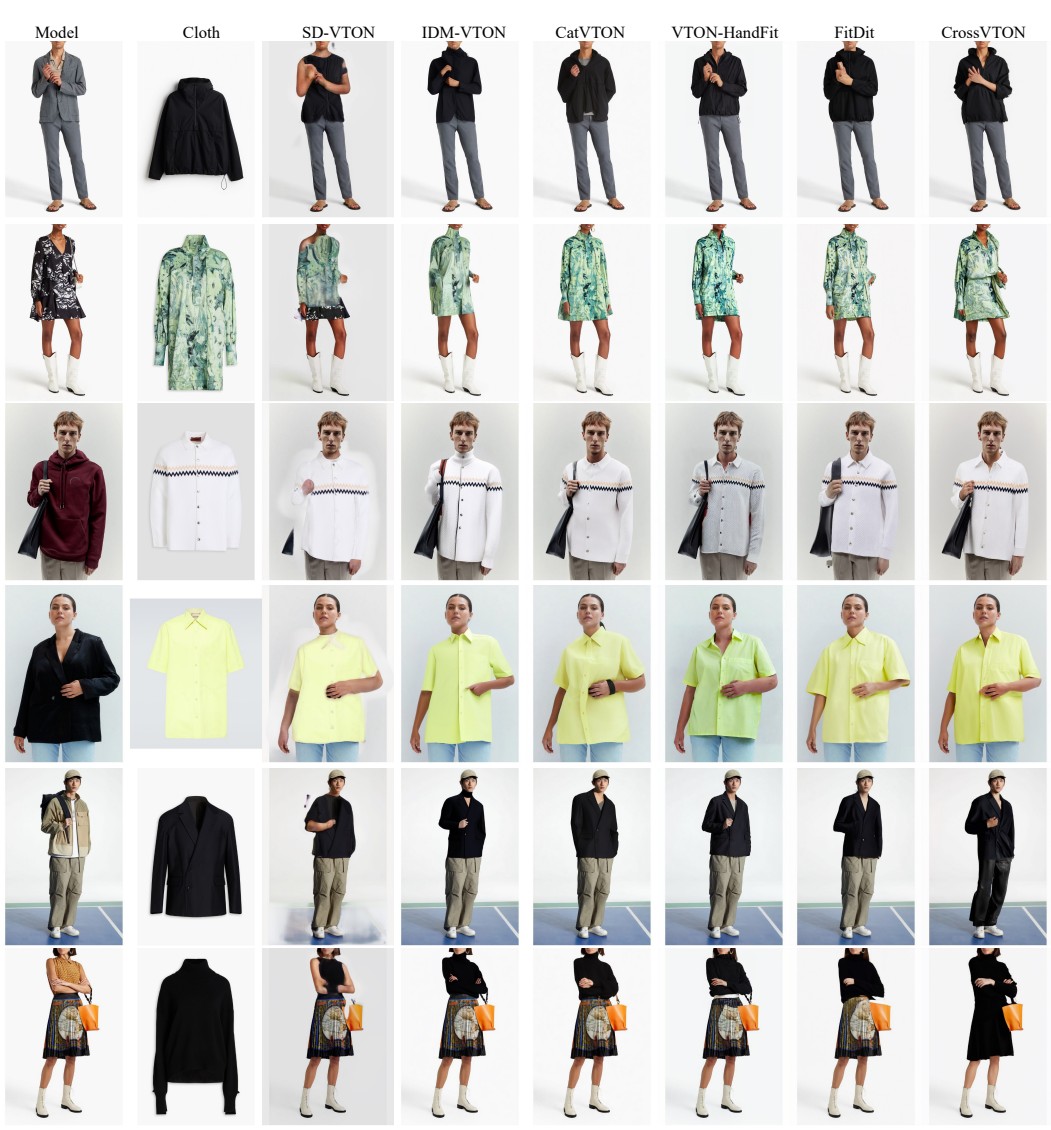

Figure 13: Visual results on Hand-Occluded Consistency Dataset (HOC) to analyze the hand-reconstruction ability for virtual try-on tasks.

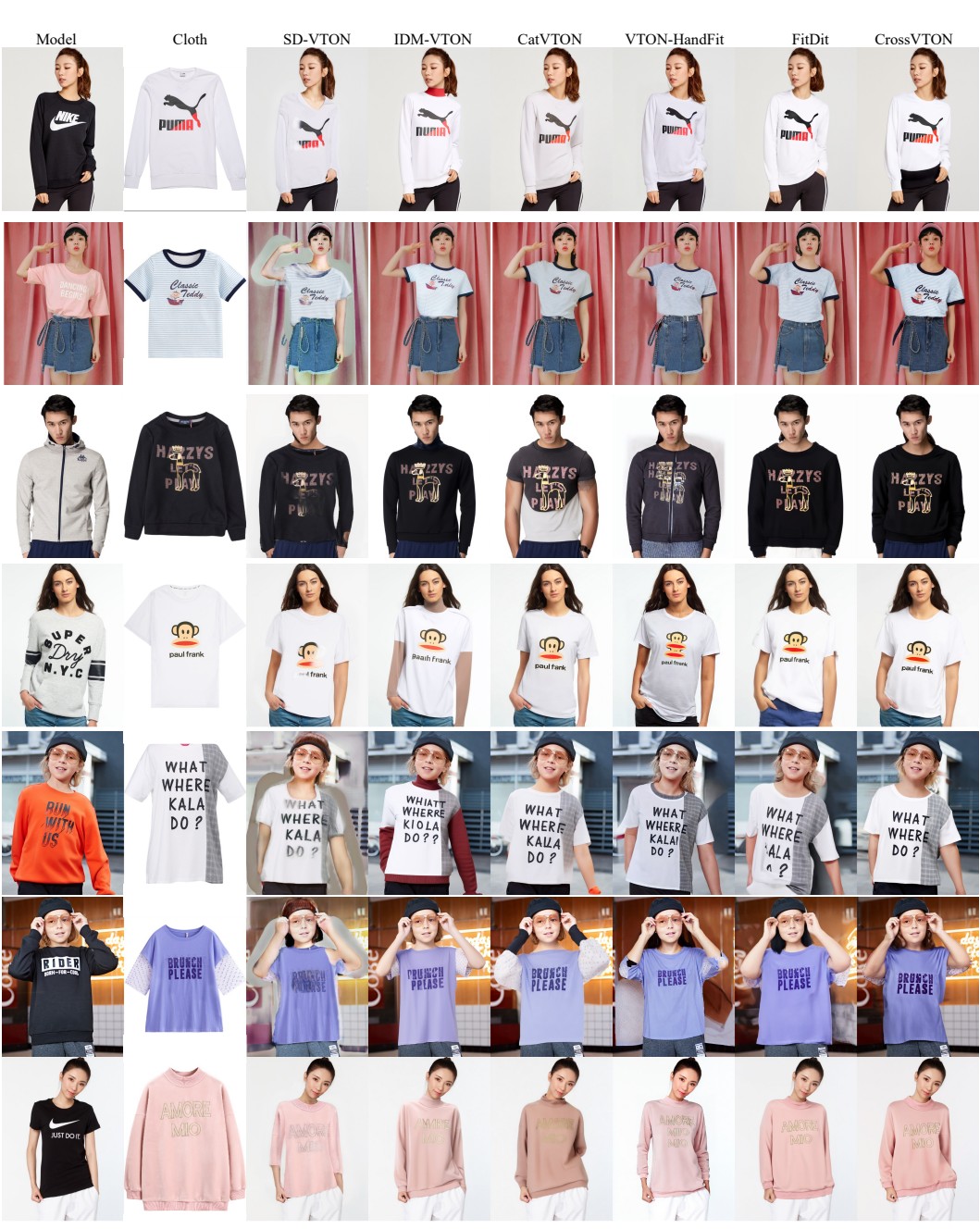

Figure 14: Visual results on Font Texture Fidelity Dataset (FTF) to analyze the rich texture-aware maintenance ability for virtual try-on tasks.

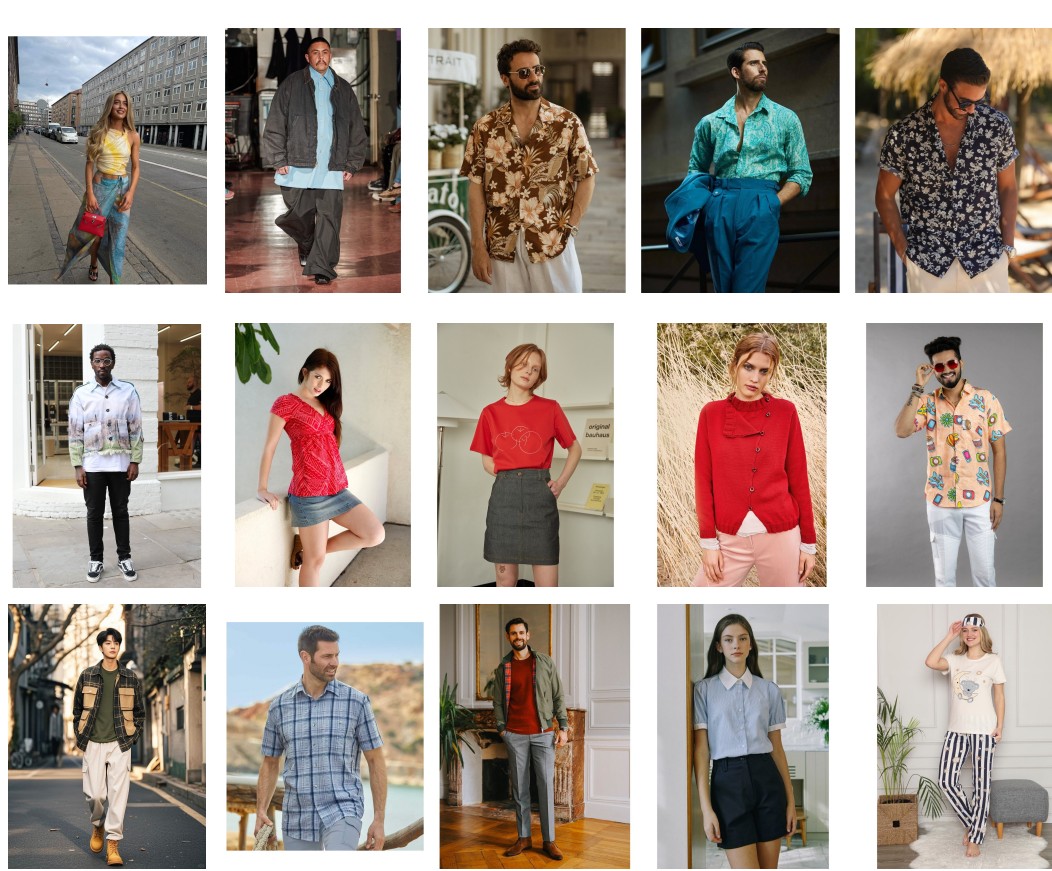

Figure 15: The visual model image samplings of Complex Background Consistency Dataset (CBC).

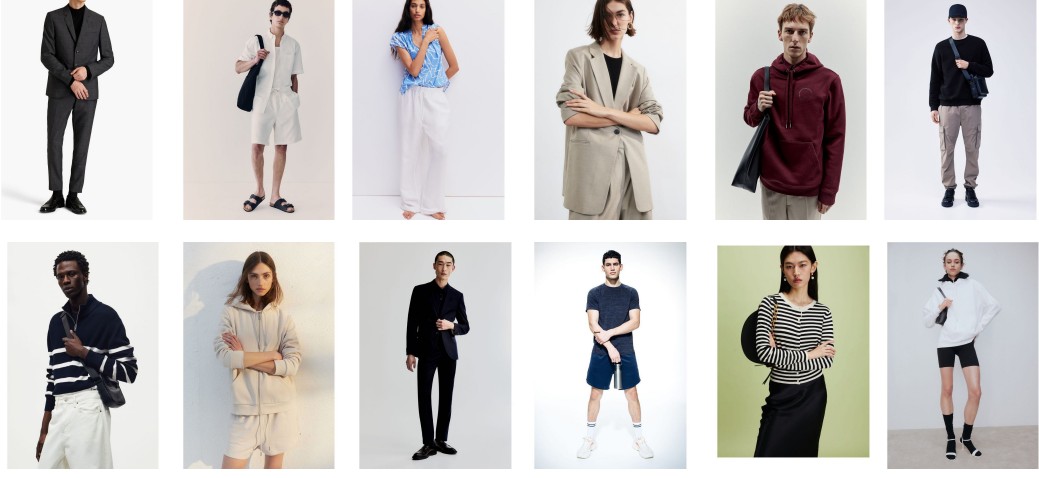

Figure 16: The visual model image samplings of Hand-occluded Consistency Dataset (HOC).

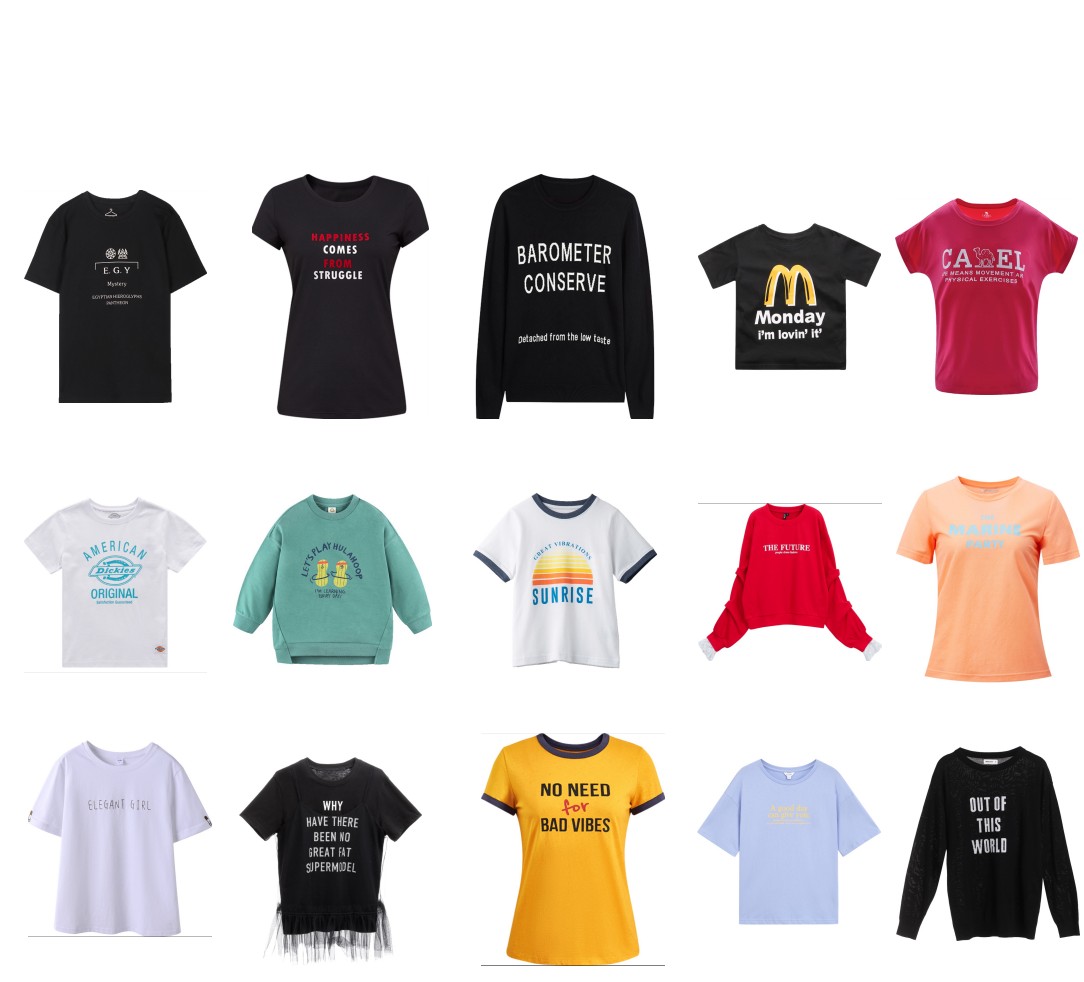

Figure 17: The visual garment image samplings of Font Texture Fidelity Dataset (FTF).

