# OpenReview forum: "VTBench: Comprehensive Benchmark Suite Towards Real-World Virtual Try-on Models"
_ICLR.cc/2026/Conference — Submitted to ICLR 2026_

### Official Review · Reviewer_Nr7j · 2025-10-22

**Soundness:** 3
**Presentation:** 3
**Contribution:** 3
**Rating:** 6
**Confidence:** 4

**Summary:**

This paper presents a hierachical virtual try-on benchmark for virtual try-on evaluation on multple aspects, including both the overall and localized texture quality of generated images. The authors use CLIP, DINO and QWen models to construct specific metrics for each aspect and address the image evaluation under the unpaired setting. The proposed benchmark also includes human preference labels to test the metrics' perceptual alignment with human.

**Strengths:**

1) The motivation of this paper precisely targets the major challenge in virtual try-on evaluation: coarse similarity metrics that are inconsistent with human perception and unsuitable for texture detail evaluation.

2) Font texture similarity is a novel metric that has been overlooked in prior work, but is very important in real-world setting where it's necessary to preserve brand's text logo.

3) The proposed benchmark is valuable to the virtual try-on community.

**Weaknesses:**

1) Using VLM model to determine size fitness is not convincing. It is difficult to evaluate if a generated garment fits the original body shape because of clothing-body occlusion in the clothed model image. In addition, size fitness itself can also be decomposed into three categories: oversized/loose fitting, normal fitting and tight fitting. The authors provide human alignment scores in the experiments, but it's better to see more evidence showing that the size evaluation in VLM is accurate. Perhaps some visualizations on randomly-selected triplets and their QWen responses can be helpful.

2) The hand consistency metric evaluates hand/joint structure but does not include various hand artifacts that are common in many virtual try-on images. I would suggest changing the name to avoid confusion or including hand artifacts detection (e.g., skin color incosistency and blurry finger edge).

3) I suggest the author provide more details of the data filtering criteria and human evaluation, including sources and number of human annotators.

**Questions:**

1) When evaluating background semantic consistency, what's the justification of choosing DINO instead of QWen VLM?

2) How is the overall rank calculated in Table 1? I assume it is not a simple average since the six metrics all have different magnitudes.

3) The sentence at L417 is cut off.

4) Figure 3 and Figure 5  show results on five dimensions. I suggest the authors also include results of fidelity. It is important as it serves as a baseline to show how mismatched FID/KID is with human judgment.

**Details Of Ethics Concerns:**

The proposed dataset contains human subjects.

---

> ### Author Response · Authors · 2025-11-23
> **Response to Reviewer Nr7j [1/2]**
>
> We greatly appreciate your insightful and detailed feedback. Your comments have been carefully discussed and fully taken into account, and we improve our work accordingly.
>
> ### **Addressing Weakness**
>
> **W1:** VLM-based size fitness evaluation lacks convincing evidence due to clothing-body occlusion and unaddressed fit granularity (oversized/normal/tight). Request: visual examples of random triplets with Qwen-VLM responses to validate accuracy beyond human alignment scores.
>
> **R1:** Thank you for the careful comment. We clarify a misunderstanding arising from our terminology. In our benchmark, ''*Size-fitting*'' does not refer to physical fit granularity (e.g., oversized/tight)  or rely on occlusion analysis.  Instead, it evaluates the semantic plausibility of cross-category try-on (e.g., long skirts ↔ upper jackets). To eliminate this ambiguity, we rename this metric to ***Cross-Category  Plausibility*** in the revised manuscript.
>
> **Regarding the evidence:** As requested, Fig. 5 presents randomly selected triplets and the corresponding QWen responses. Specifically, "*Accepted*" and "*Rejected*" denote the model's assessment of the logical plausibility (or implausibility) of the cross-category try-on. These examples demonstrate that the VLM accurately judges the logical compatibility of source-target category pairings rather than physical dimensions.
>
> **W2:** Hand consistency only evaluates hand and joint structure, but not common hand artifacts (e.g., skin-tone inconsistency or blurry finger edges). I recommend renaming the metric or incorporating artifact detection to avoid confusion.
>
> **R2:** Thank you for your suggestions. To improve clarity, we rename this metric to hand-structure consistency in the revised manuscript. In addition, we add an evaluation metric for hand skin tone. As shown in Fig. 4, the hand skin tone metric aligns well with human preference annotations (ρ > 0.8). We add this metric to the revised manuscript.
>
> **W3:** I suggest the author provide more details of the data filtering criteria and human evaluation, including sources and the number of human annotators.
>
> **R3:** We appreciate the suggestion to elaborate on our experimental protocols. We clarify the specific details below and will incorporate them into the camera-ready version:
>
> **(1) Data Filtering Criteria:** We employ rigorous manual curation to ensure high-quality benchmarks. Specifically:
>
> - **Hand:** Selects frontal portraits featuring explicit hand-garment occlusion.
>
> - **Texture:** Focuses on garments containing text or dense patterns.
>
> - **Cross-Category:** Enforces mismatched categories between the source garment and the target person.
>
> - **Complex-Background:** Consists of natural scene backgrounds rather than simple studio backdrops.
>
> **(2) Data sources:** To ensure diversity and realism, we aggregate images from a wide range of public sources, primarily including brand retail homepages and e-commerce platforms.
>
> **(3) Human Evaluation:** The study involves 20 participants. Crucially, all evaluators receive explicit training on the assessment criteria prior to the study to ensure consistency and reliability.

---

> ### Author Response · Authors · 2025-11-23
> **Response to Reviewer Nr7j [2/2]**
>
> ### **Response to Questions**
>
> **Q1:** When evaluating background semantic consistency, what's the justification of choosing DINO instead of QWen VLM?
>
> **R1:** We select DINO based on the specific nature of the background consistency task. Background variations in VTON are typically confined to narrow regions surrounding the human subject (e.g., inpainting artifacts). According to [1], DINO is highly effective at capturing local structural patterns and feature correspondence in these specific areas. In contrast, Qwen-VLM focuses on global semantic descriptions, which often lack the spatial sensitivity required to detect subtle background deformations around the garment-body boundary.
>
> **Q2:** How is the overall rank calculated in Table 1? I assume it is not a simple average since the six metrics all have different magnitudes.
>
> **R2:** We introduce the ranking strategy inspired by the Friedman test [2, 3] to address the scale differences across metrics. Specifically, we first calculate the rank of each method independently for every individual metric. Then, we average these individual ranks to derive the final Overall Rank. This approach ensures that the final ranking is not biased by the varying magnitudes of different metrics.
>
> **References:**
>
> [1] Dinov2 meets text: A unified framework for image-and pixel-level vision-language alignment. CVPR2025.
>
> [2] Milton Friedman. The Use of Ranks to Avoid the Assumption of Normality Implicit in the Analysis of Variance. *Journal of the american statistical association*, 1937.
>
> [3]  Milton Friedman. A comparison of alternative tests of significance for the problem of m rankings. *The annals of mathematical statistics*, 1940
>
> **Q3:** The sentence at L417 is cut off.
>
> **R3:** We apologize for this formatting oversight. The sentence at L417 is obscured by Fig. 3. The complete sentence reads: “*...suggesting that existing aesthetic metrics are inadequate for unpaired VTON tasks.*” We have corrected it in the revised manuscript.
>
> **Q4:** Figure 3 and Figure 5 show results on five dimensions. I suggest the authors also include results of fidelity. It is important as it serves as a baseline to show how mismatched FID/KID is with human judgment.
>
> **R4:** We appreciate this insightful suggestion. To demonstrate the misalignment between fidelity metrics and human perception, we analyze the correlation of FID/KID with human ratings. Given that FID and KID are distribution-based metrics and cannot be computed for individual images, standard correlation analysis is inapplicable. Consequently, we adopt an alternative validation strategy. Specifically, we divide the generated results into "Good" and "Bad" groups based on human preference annotations and compute FID/KID for each group against the real data distribution. As illustrated in Fig. 6, the FID/KID scores exhibit a counter-intuitive trend: the 'good' group yields higher (worse) scores than the 'bad' group. This quantitative analysis demonstrates that generic fidelity metrics do not reliably reflect human perceptual judgments in VTON tasks.

---

### Official Review · Reviewer_bco4 · 2025-10-30

**Soundness:** 4
**Presentation:** 4
**Contribution:** 3
**Rating:** 8
**Confidence:** 5

**Summary:**

This paper proposes VTBENCH , a comprehensive benchmark suite for evaluating image-based virtual try-on (VTON) models. The authors argue that existing evaluation methods (like FID/KID) align poorly with human perception and lack consideration for real-world complex scenarios. VTBENCH addresses these issues with a hierarchical evaluation framework that decomposes VTON quality into three main categories and six dimensions: General Image Quality (Similarity, Aesthetics), Garment Preservation (Texture, Size), and Auxiliary Consistency (Background, Hand) .

**Strengths:**

- The paper's most outstanding strength is its large-scale human preference annotation study. The results  robustly demonstrate that the proposed new metrics (e.g., for cross-category, texture, hand, and background consistency) are highly correlated with human perceptual judgments , a feature broadly lacking in existing metrics. The VLM-based "Size Fitness" metric and the OCR-based "Font Texture Similarity" (FTS) metric  are highly innovative. They elevate evaluation from low-level pixel similarity to high-level semantic and logical correctness.

- The curation of four new specialized test sets (CBC, FTF, CSF, HOC) is a significant contribution, specifically targeting common failure cases like hand occlusions , complex backgrounds , and cross-category try-ons  that are ignored by previous benchmarks.

- The comprehensive benchmark analysis of 16 SOTA models provides valuable insights to the community regarding the pros and cons of different architectures (GAN vs. UNet vs. DiT) .

**Weaknesses:**

- **Questionable Efficacy of Visual Texture Metric**: The paper computes the cosine similarity between CLIP or DINO embeddings of the original garment and the cropped garment region from the generated image  to judge visual texture. However, CLIP and DINO are trained via contrastive or self-supervised learning, which are not inherently designed to enhance fine-grained details. For example, generative methods like IP-Adapter, which use CLIP as an image encoder, often fail to restore reference image details. Furthermore, AnyDoor uses DINO as an encoder but still requires a high-frequency filter to ensure detail consistency. This casts doubt on the ability of CLIP or DINO to reliably measure fine-grained texture.
- **High Dependency of Background Consistency Metric on Masks**: In the measurement of background consistency , most try-on methods are mask-based, and the masked regions differ between methods. The fit and extent of a given mask will heavily influence the paper's background consistency metric. When evaluating the 16 baselines on the CBC dataset, was a standardized, pre-computed mask provided to all models?
- **Limited Scope of Hand Consistency Metric**: The Hand Consistency metric  focuses on the consistency of the model's preserved regions, but focusing only on hands is somewhat limited. Since try-on images are often half- or full-body, the hand region has a low pixel ratio. Other elements like the model's hair, face, skin tone, and body shape are visually more critical to the realism of the result and should also be evaluated for consistency.

**Questions:**

- When evaluating the 16 baselines on the CBC dataset, was a standardized, pre-computed mask provided to all models? If so, how was this mask generated? If not (i.e., each model used its own internal mask), how does this metric fairly compare different models (as the score might reflect the mask's size more than the background preservation quality)?

- Why did the authors choose to focus exclusively on hand consistency? Are there plans to expand this consistency evaluation to other non-edit regions that are critical for perceptual realism, such as facial fidelity, hair details, and skin tone consistency?

- Given the known limitations of CLIP/DINO in capturing high-frequency details (as noted in the "Weaknesses" section), can the authors provide more evidence that $E_{\epsilon}$  reliably measures fine-grained texture (rather than just style or the overall shape)? Have the authors considered supplementing this with other detail-focused metrics (e.g., high-frequency variants of LPIPS or SigLIP which is used in FLUX Redux) to enhance the "Texture Fidelity" dimension's evaluation?

---

> ### Author Response · Authors · 2025-11-23
> **Response to Reviewer bco4**
>
> We greatly appreciate your insightful and detailed feedback. Your comments have been carefully discussed and fully taken into account, and we improve our work accordingly.
>
> ### **Response to Questions**
>
> **Q1:** Was a standardized mask used for all models on CBC dataset? If not, isn’t the metric biased by mask differences rather than background preservation quality?
>
> **R1:** We clarify that our evaluation decouples the inference mask from the evaluation mask to ensure fairness. First, when generating visual try-on results by different methods, we allow each method to use its inherent masking strategy (treating it as a crucial part of each method). Second, when computing the background consistency metric, we employ the same off-the-shelf segmentation algorithm to segment the background region for all methods. Consequently, our metric evaluates the consistency between the generated and original backgrounds within this standardized region. Finally, since we calculate the average pixel-wise difference, the metric reflects per-pixel fidelity and is robust to variations in mask size, ensuring fair comparison regardless of mask dimensions.
>
> **Q2:** Why focus only on hands? Will evaluation expand to other critical non-edited regions like face, hair, and skin tone?
>
> **R2:** (1) Hand regions are prioritized due to their high incidence of occlusion artifacts and structural distortions in virtual try-on. In contrast, facial and hair regions demonstrate consistently high fidelity in our validation set, rendering them lower priority for initial metric design.
>
> (2) Thank you for your suggestion. Expanding consistency evaluation to regions like face, hair, and skin tone significantly enhances the completeness of the benchmark. We plan to incorporate face and hair metrics in future versions. However, given the time constraints of the rebuttal period, we focus our immediate analysis on hand skin tone consistency using the existing dataset. Specifically, we conduct human preference annotations and correlation analysis for this metric. As shown in **Fig. 4**, our findings confirm a strong alignment between the hand skin tone metric and human preference (ρ>0.85). We add this correlation analysis to the revised manuscript.
>
> **Q3:** Given CLIP/DINO's limitations in high-frequency details, should high-frequency variants of LPIPS or SigLIP supplement texture fidelity evaluation?
>
> **R3: (1) Rationale for CLIP & DINO:** In [1], DINOv2 is described as follows: “This self-supervised model, trained to capture both the global context and local information of the image, has led to state-of-the-art performance in tasks that require an overall understanding of the image such as classification and those that necessitate more **fine-grained details** such as segmentation……” So we believe DINO can provide low-level fine-grained details. While CLIP focuses on high-level semantics. Thus, the CLIP-DINO combination effectively provides both semantic and vision details when evaluating texture fidelity.
>
> **(2) Evaluation of LPIPS & SigLIP:** We evaluate LPIPS and SigLIP against human preference annotations on our texture dataset. As shown in Fig. 4, LPIPS exhibits weak alignment with human judgments (ρ < 0.4), demonstrating its inadequacy for texture preservation assessment. In contrast, SigLIP achieves significantly higher correlation (ρ > 0.8). We incorporate the correlation analysis of SigLIP and LPIPS into the revised manuscript to strengthen the evaluation of texture fidelity.
>
> **References:**
>
> [1] Dinov2 meets text: A unified framework for image-and pixel-level vision-language alignment. CVPR2025.

---

> > ### Comment · Reviewer_bco4 · 2025-11-24
> > **Response to Authors**
> >
> > Thank the authors for the response.
> >
> > Regarding the discussion on DINO and CLIP metrics, the provided results suggest that SigLIP appears to be more suitable than DINO for evaluating fine-grained details. Without experimental evidence, the claims drawn from the original DINO paper regarding its ability to evaluate clothing textures are not sufficiently convincing. Therefore, I maintain my original viewpoint, though I appreciate the additional evaluation results of LPIPS and SigLIP.
> >
> > Regarding the issue of mask usage in background evaluation, most of the evaluated methods are trained on public datasets such as VITON-HD, which contain almost no complex backgrounds. Furthermore, the use of different masking strategies by different methods indeed leads to significant discrepancies in this evaluation. However, since background evaluation is relevant for future mask-free (image editing-based) virtual try-on methods, this section remains valuable for VTON research.
> >
> > Additionally, while reviewing the revised manuscript, I noticed another potential issue. Figure 10 is used to compare Cross-Category Plausibility (CCP); however, the first four methods in rows 1-2 do not appear to utilize the correct masks (e.g., when changing a dress, the pants were not masked/covered). This could result in an unfair comparison.
> >
> > Regarding my rating of 8: Given that this is a benchmark-oriented paper, the evaluation framework and experiments are comprehensive and sufficient, and the writing is clear and accessible. Despite some existing weaknesses, it remains a high-quality contribution. Based on the authors' response, I will maintain this score.

---

### Official Review · Reviewer_ALsj · 2025-10-30

**Soundness:** 3
**Presentation:** 2
**Contribution:** 3
**Rating:** 4
**Confidence:** 5

**Summary:**

This paper introduces VTBench, a benchmark for evaluating virtual try-on (VTON) models. The key contributions are three-fold: 1 A hierarchical evaluation framework that decomposes virtual try-on quality into six fine-grained dimensions; 2 It introduces several unpaired evaluators to overcome the lack of paired ground-truth images; 3 It provides four custom test datasets with human preference annotations, where a model comparison study is conducted to evaluate different VTON models.

**Strengths:**

+ This work establishes a foundation for future research toward realistic and perceptually aligned virtual try-on systems.
+ It evaluates 16 state-of-the-art models across multiple paradigms, offering valuable comparative insights for the community.
+ It tailors existing models for virtual try-on evaluation from different perspectives.

**Weaknesses:**

- The aesthetic metric correlates poorly with humans, significantly discrediting the reliability of its use in evaluating and comparing VTON models.
- It lacks comparison with commonly used full-reference and no-reference image quality assessment metrics.
- Its claim to guide the development of future VTON models is somewhat overstated, since the benchmark itself doesn’t propose novel generative methods or optimization strategies.

**Questions:**

N/A

---

> ### Author Response · Authors · 2025-11-23
> **Response to Reviewer ALsj**
>
> # Addressing Weakness
>
> We greatly appreciate your insightful and detailed feedback. Your comments have been carefully discussed and fully taken into account, and we improve our work accordingly.
>
> **W1:** The aesthetic metric correlates poorly with humans, significantly discrediting the reliability of its use in evaluating and comparing VTON models.
>
> **R1:** We sincerely understand your concern that aesthetic metric correlates poorly with humans. As discussed in **Appendix A.3** (line 752), we also note its limited correlation with human judgments. However, we retain this metric for the following reasons:
>
> **1. Completeness & Benchmarking:** The aesthetic metric serves as a standard measure in generative modeling (similar to FID and KID) to assess general image quality, ensuring fair comparison with prior works.
>
> **2. Task-Specific Focus:** Our evaluation focuses on *Garment Preservation* and *Auxiliary Consistency*. These task-specific metrics align strongly with perceptual fidelity and structural plausibility, as supported by human annotations.
>
> To further clarify this, we will incorporate relative importance weights for all metrics in the camera-ready version of VTBench to guide users toward more reliable indicators.
>
> **W2:** It lacks comparison with commonly used full-reference and no-reference image quality assessment metrics.
>
> **R2:** **(1) Full-Reference (FR) Metrics:** Metrics like PSNR, SSIM, and MSE strictly require pixel-aligned ground truth, which typically correspond to evaluating the same garment swapped onto a garment-masked person. However, practical VTON is inherently an unpaired generation task, where the target garment must accommodate different poses and body shapes. Since there is no pixel-perfect "ground truth" for these structural deformations, calculating FR metrics is generally unsuitable. Therefore, we focus on unpaired evaluation, which better reflects real use cases and provides more meaningful comparisons across methods.
>
> **(2) No-Reference (NR) Metrics:** We investigate two representative NR metrics (blur and noise estimation) to assess texture quality. However, as illustrated in Fig. 4, we observe an extremely weak correlation (ρ < 0.25) between these metrics and human preference. This aligns with the consensus in the generative modeling community that low-level IQA metrics often fail to capture the detailed texture feature required for VTON. Consequently, we focus on task-specific evaluations that better reflect user experience.
>
> **W3:** Its claim to guide the development of future VTON models is somewhat overstated, since the benchmark itself doesn’t propose novel generative methods or optimization strategies
>
> **R3:**  We claim that VTBench is a benchmark framework rather than a specific method. However, we maintain that rigorous evaluation is the prerequisite for algorithmic advancement. We believe VTBench guides the optimization direction of future VTON research from two critical perspectives:
>
> **(1) Evaluation drives innovation:** Current methods predominantly rely on generic metrics (e.g., FID/KID) for optimization. However, our analysis reveals that these metrics fail to capture task-specific nuances such as garment structure, hand consistency, and background integrity. VTBench fills this gap by providing explicit optimization objectives:
>
> - **Hand-Structure Consistency** encourages models to minimize anatomical distortions.
>
> - **Texture Preservation** guides models to retain detailed texture features.
>
> - **Cross-Category Plausibility** guides models to understand the logical plausibility of cross-category try-on.
>
> - **Background Consistency** ensures the consistency of complex scenes before and after visual try-on.
>
>   By optimizing against these specific criteria rather than generic distribution metrics, future models can achieve higher practicality and robustness.
>
> **(2) Data and structure for rigorous benchmarking:** VTBench establishes a high-quality test set derived from an initial pool of **50,000** real-world images sourced from online retail platforms. Through rigorous manual curation, we construct four specialized datasets targeting distinct failure modes, including hand occlusion, cross-category plausibility, texture fidelity, and background consistency. Consequently, VTBench provides a comprehensive and standardized evaluation suite. It features a substantial test set covering **6 core evaluation dimensions and 14 distinct metrics**. Crucially, each dimension is grounded in **human preference annotations**. This structured framework enables researchers to directly utilize VTBench for precise, multi-faceted assessments. It allows for the diagnosis of model limitations beyond simple aggregate scores, ensuring an accurate and holistic evaluation of VTON performance.

---

> ### Comment · Reviewer_ALsj · 2025-11-24
>
> Thanks for the reponses.
>
> Regarding W1: The authors emphasize that the evaluation focuses on Garment Preservation and Auxiliary Consistency. However, this is inconsistently with descriptions in the manuscript. In Sec. 3.1, the authors claim that **three main perspectives** are considered: 1) General Image Quality 2）Garment Preservation and 3) Auxilliary Consistency, suggesting that  General Image Quality is equally important as Garment Preservation and Auxilliary Consistency. It is unfair to change the claim simply due to the poor correlation results of the employed metrics.
>
> Regarding W2: Please specify the blur and noise estimation metrics in more details with appropriate reference. In addition, I strongly recommend the authors to try advanced NR-IQA metrics, especially VLM-based or MLLM-based ones. For example, VisualQuality-R1.
>
> Regarding W3: A benchmark should be accompanied with reliable quantiative metrics. Given the poor correlation of the employed aesthetic score, this provide limited insight on guiding furture development of virtual try-on methods to improve general image quality.

---

> ### Author Response · Authors · 2025-11-29
> **Response to Reviewer ALsj [1/2]**
>
> Thanks for your suggestions.
> 1. As you point out, General Image Quality constitutes a critical dimension of VTBench. We acknowledge that the low correlation with human preference indicates that the LAION aesthetic model is not suitable for evaluating the aesthetic quality of virtual try-on results. We therefore evaluate several alternative aesthetic scoring models, including HumanAesExpert [1] and AesExpert [2]. However, HumanAesExpert shows a low correlation (ρ=0.14) with subjective human preference and AesExpert does not provide numerical aesthetic scores (only generating textual descriptions). As shown in Fig. 11, HumanAesExpert fails to detect specific defects such as blurred regions and artifacts,  and often assigns high scores to images with obvious flaws. To address this, we further experiment with Qwen3-VL and design prompts specifically for aesthetic evaluation to better target these fundamental distortion indicators. This model achieves a high correlation with human preference (ρ=0.89) and demonstrates strong aesthetic evaluation capability. Therefore, we replace LAION with Qwen3-VL as our aesthetic evaluation model.  We thank you for the suggestion, which has significantly improved the accuracy of aesthetic assessment of VTBench.
>
> 2. We also conduct a comprehensive evaluation of VisualQuality-R1 [3] across various dimensions of the test set. Its correlations with human preference in aesthetics, background, and texture are 0.75, 0.71, and 0.77, respectively. Although these are slightly lower than our specialized metrics (0.89, 0.83, and 0.87, respectively), they indicate that VisualQuality-R1 effectively reflects overall image quality. However, correlations are lower in try-on specific dimensions such as cross-category and hand-structure (0.35 and 0.55, respectively). Therefore, we consider VisualQuality-R1 suitable as part of the General Image Quality dimension to capture holistic quality.
>
> 3. Inspired by your suggestions, we refine the General Image Quality dimension by adopting Qwen3-VL for aesthetic evaluation and incorporating VisualQuality-R1 to assess overall image quality. Together with the Garment Preservation and Auxiliary Consistency metrics, this forms a more comprehensive and accurate evaluation framework. With the support of our carefully annotated dataset, we believe VTBench will provide more reliable guidance for researchers in this field.
>
> [1] HumanAesExpert: Advancing a Multi-Modality Foundation Model for Human Image Aesthetic Assessment. Arxiv, 2025.
>
> [2] AesExpert: Towards Multi-modality Foundation Model for Image Aesthetics Perception. ACM MM 2024.
>
> [3] VisualQuality-R1: Reasoning-Induced Image Quality Assessment via Reinforcement Learning to Rank. NeurIPS 2025.

---

> ### Author Response · Authors · 2025-11-29
> **Response to Reviewer ALsj [2/2]**
>
> We appreciate you pointing out the omission of details regarding the blur and noise estimation metrics. The specific implementation details are as follows:
>
> Blur Metric: We utilize the Variance of the Laplacian, a standard operator for focus measurement. This method calculates the variance of the second derivative of the image, where high variance indicates sharp edges. It is widely adopted in shape-from-focus tasks and image quality analysis [1].
>
> Noise Metric: We adopt a fast noise variance estimation method based on residual analysis in the spatial domain. According to [2], this approach involves filtering the image (to suppress texture) and calculating the standard deviation of the residuals in homogeneous regions to robustly estimate additive noise levels.
>
> [1] Diatom autofocusing in brightfield microscopy: a comparative study. ICPR, 2000.
>
> [2] Fast Noise Variance Estimation. Computer Vision and Image Understanding, 1996.

---

### Author Response · Authors · 2025-11-23
**General Response**

We highly value the constructive and meaningful feedback from all reviewers. Having carefully considered their comments, we revise our submission accordingly in the latest version. To summarize, we conduct following supplementary experiments:

**1. Hand Skin Tone Metric:** For hand skin tone consistency, we collect an additional set of human preference annotations. We validate the alignment between the hand skin tone metric and these human judgments in **Fig. 4**.

**2. Validation of Detail–Texture Metrics with Human Preference:** We present the consistency results between detail–texture metrics (LPIPS and SigLIP) and human preference in **Fig. 4**.

**3. Validation of No-Reference Metrics with Human Preference:** We evaluate the alignment between additional no-reference metrics and human preference, including two representative indicators (blur and noise estimation), as shown in **Fig. 4**.

**4. Accepted and Rejected VLM Response in Cross-Category VTON:** We present randomly selected qualitative examples demonstrating both accepted and rejected cases by the VLM in **Fig. 5**. "Accept" and "Rejected" denote the model's assessment of the logical plausibility (or implausibility) of the cross-category try-on. These samples confirm the model's discriminative capability in cross-category scenarios.

**5. FID/KID Comparison “Good vs Bad” Human Ratings:** We analyze the consistency of FID and KID with human ratings by dividing results into "Good" and "Bad" groups in **Appendix Fig. 6**.

The modifications are colored in blue in the latest version of the manuscript. We sincerely hope our responses have effectively addressed the concerns raised by the reviewers and clarified any potential confusion, and we are happy to engage in further discussions if needed.

---

### Meta-Review · Program_Chairs · 2026-01-03

**Summary:**

The reviewer were primarily concerned with:
1. The limited scope of the hand metrics
2. Poor correlation of the aesthetics metric
3. Efficacy of the VLM automated evaluation

To partially address these concerns, the authors have demonstrated reasonably high correlation of individual metrics with respect to the gold standard human preference by incorporating suggestions from individual reviewers.

Apart from these issues, this paper uses many images scraped from online retail platforms. For example, here is an image that appears in the paper itself that is from the H&M catalog: https://id.hm.com/id_en/slim-fit-single-breasted-jacket-0713986070.html The paper claims that "All data used in this work are collected exclusively from public domains and open-licensed sources", but this is clearly not the case. For example, H&M's terms and conditions says "Any other use, including but not limited to commercial purposes, is strictly prohibited without prior written consent from H&M Group and the respective copyright owner(s)." The authors need to address these legal and licensing issues in order for this dataset to be published.

**Reviewer Concerns:**

Most correlation metrics and additional metric suggestions have been addressed. The limited focus on the hand metric remains but appears to still provide a relatively high correlation. Reviewers' ethical concerns have not been addressed.

**Reviewer Scores:**

I think it is unlikely the reviewers scores would have changed significantly.

---

### Decision · Program_Chairs · 2026-01-26

Reject